# Region-specific denoising identifies spatial co-expression patterns and intra-tissue heterogeneity in spatially resolved transcriptomics data

Linhua Wang [1], Mirjana Maletic-Savatic [2,3] & Zhandong Liu [2,3] ✉

Spatially resolved transcriptomics is a relatively new technique that maps transcriptional information within a tissue. Analysis of these datasets is challenging because gene expression values are highly sparse due to dropout events, and there is a lack of tools to facilitate in silico detection and annotation of regions based on their molecular content. Therefore, we develop a computational tool for detecting molecular regions and region-based Missing value Imputation for Spatially Transcriptomics (MIST). We validate MIST-identified regions across multiple datasets produced by 10x Visium Spatial Transcriptomics, using manually annotated histological images as references. We benchmark MIST against a spatial k-nearest neighboring baseline and other imputation methods designed for single-cell RNA sequencing. We use holdout experiments to demonstrate that MIST accurately recovers spatial transcriptomics missing values. MIST facilitates identifying intra-tissue heterogeneity and recovering spatial gene-gene co-expression signals. Using MIST before downstream analysis thus provides unbiased region detections to facilitate annotations with the associated functional analyses and produces accurately denoised spatial gene expression profiles.

Both healthy development and pathogenic processes involve changes in the expression patterns of numerous genes. Both processes also alter the organization of cells within tissues, which has prompted interest in acquiring more granular detail to understand transcriptomic profiles of cells located throughout the two-dimensional geography of a given tissue. This approach, known as spatial transcriptomics[1,2], has been used to map gene expression across the brain and other tissues, and many tumor types[3–8]. Although various techniques have been developed to map gene expression in a way that accounts for spatial location within the tissue[9], 10X Visium Spatial Transcriptomics (ST) is the most widely used because of its whole-genome scalability, cost-efficiency, and comparative ease of use[2,8].

There are two main challenges in analyzing data produced by ST. First, although ST assigns the location of every sequenced tissue domain (spot), it does not provide region assignments based on molecular contents. Researchers typically mark anatomical regions manually by looking at the histological image aligned with ST, but some biologically distinct regions are not detectable by the eye. Moreover, assigning each spot to a region is labor-intense and provides many opportunities for error. Some computational methods have sought to assign every spot to a cluster, even when that spot resides at the boundary between pathogenically different regions[10–12]. However, these spots at the boundaries might be cell type admixtures due to the lack of single-cell resolution of spatial spots, thus not

[1]Graduate School of Biomedical Sciences, Program in Quantitative and Computational Biosciences, Baylor College of Medicine, Houston, TX, USA. [2]Jan and Dan Duncan Neurological Research Institute at Texas Children's Hospital, Houston, TX, USA. [3]Department of Pediatrics, Baylor College of Medicine, Houston, TX, USA. ✉e-mail: zhandonl@bcm.edu

belonging to a specific molecular region and should be excluded from cluster-based analysis downstream. Unbiased region mapping within ST tissues that excludes potential admixture spots would therefore present a substantial improvement in accuracy of ST data interpretation.

The second major challenge in analyzing ST data is that technical dropouts can make the gene expression profile sparse and produce excessive zero values in the gene expression data[8]. The result is a drastic reduction in ST transcriptional signals and inaccurate co-expression calculations, cluster detection, and other downstream analyses. Therefore, computational methods designed explicitly to denoise the dropped-out expression values in ST are needed.

To address these two challenges, we developed a computational tool called Missing-value Imputation and in silico region detection for Spatially resolved Transcriptomics (MIST). MIST detects tissue regions based on their molecular content by maintaining neighboring spots that are both molecularly similar and physically adjacent. Assuming each detected molecular region has a limited number of cell types, MIST then denoises the missing values by approximating a low-rank gene expression matrix through nuclear-norm minimization algorithm[13].

## Results

### The MIST algorithm

MIST uses two steps to address the challenges in ST analyses: boundary detection and imputation (Fig. 1, Supp. Fig. 1). In the first step, MIST automatically detects molecular regions in the sample of interest. MIST embeds ST as a two-dimensional graph where every spot is represented as a node. Every pair of adjacent nodes is connected by an edge whose weight is defined by the molecular similarity of the two nodes. To simulate region boundaries, MIST filters out low-weight edges with a threshold and extracts the connected components within the remaining graph (Supp. Algorithm 1). To avoid bias in selecting the filtering threshold, MIST searches for the optimal value that maximizes average intra-region similarities, minimizes inter-region similarities, and is regulated by the proportion of spots that were left-out as isolated spots (see Methods). By doing so, MIST provides a region-specific and spot-level expression profile for the whole transcriptome, enabling an understanding of regional transcriptional differences.

In the second step, MIST estimates the missing values in each detected region by averaging the outcomes from multiple runs of a low-rank approximation algorithm[13]. Under the assumption that such a region should contain a number of cell types, we expect the denoised region-specific expression matrix to have a low-rank. We, therefore, used a low-rank matrix completion approach that estimates the missing values by minimizing the singular values of the denoised matrix (Supp. Algorithm 2).

To increase the reliability of the estimated values, MIST uses a mini-batch-based ensemble method that takes the low-rank completion as the baseline to generate multiple estimates, and then averages outcomes from individual batches to make the final predictions. To superimpose biological information in building the batches, MIST constructs every mini-batch with prior region detection information, i.e., every batch is composed with spots from a core region, some spots randomly sampled from other regions, and the isolated spots (Supp. Algorithm 3).

### MIST facilitates annotations of molecular regions in melanoma

To demonstrate MIST's efficacy in region detection and how it helps annotation, we applied it to a melanoma sample (Fig. 2a). MIST faithfully detected major molecular regions that agreed with the human histological annotation of the tumor (black region), lymphoid cells (dark orange region) and stroma (red region) (Fig. 2b, Supp. Fig. 2).

MIST detects significantly upregulated genes for each molecular region (Fig. 2d, see Methods). In the black region—annotated as a tumor region by human experts—we found 143 activated genes (Fig. 2d). Among them were several well-known tumor marker genes, including SFRP1[14], ATP1A1[15], GNAS[16], NDRG1[17], and SPP1[18]. These genes were at the top of the melanoma (black region)-activated list of genes that MIST-identified, supporting its accuracy. Thus, we can now visualize the gene expression pattern of any gene of interest with the regional boundaries circled out by MIST. For example, SFRP1, a tumor marker gene, showed elevated expression values in the black region (Fig. 2c).

Next, to understand not only molecular but also functional differences in the identified regions, MIST performs gene set enrichment analysis (GSEA) to find significantly enriched gene ontology terms. For example, the "Peptide antigen assembly with MHC protein complex" dominates in the dark orange region (FDR = $5 \times 10^{-5}$, hypergeometric test adjusted by multiple comparisons), suggesting that this is not a melanoma tumor or a stroma region, but a lymphoid region (Fig. 2e). In turn, the "Humoral immune response mediated by circulating

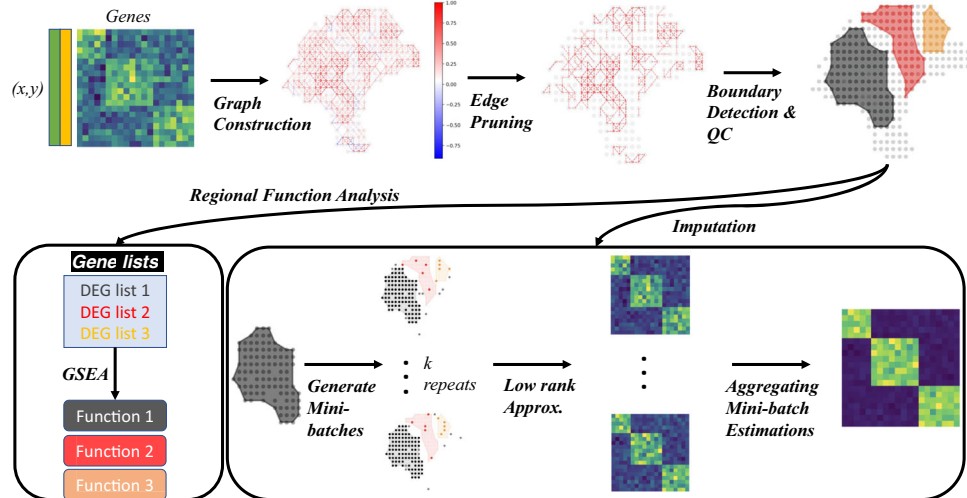

**Fig. 1 | The MIST pipeline.** MIST constructs a graph using the spatial gene expression matrix, removes edges with low weights, and draws the boundaries to detect molecular regions within the ST sample. Next, MIST enables two functionalities. The regional function analysis module detects regional activated gene lists and uses Gene Set Enrichment Analysis (GSEA) to guide functional annotation of each region. The imputation module utilizes a region-based mini-batch of low-rank approximation algorithm and estimates the final denoised expression values by averaging the outcomes from all mini-batches.

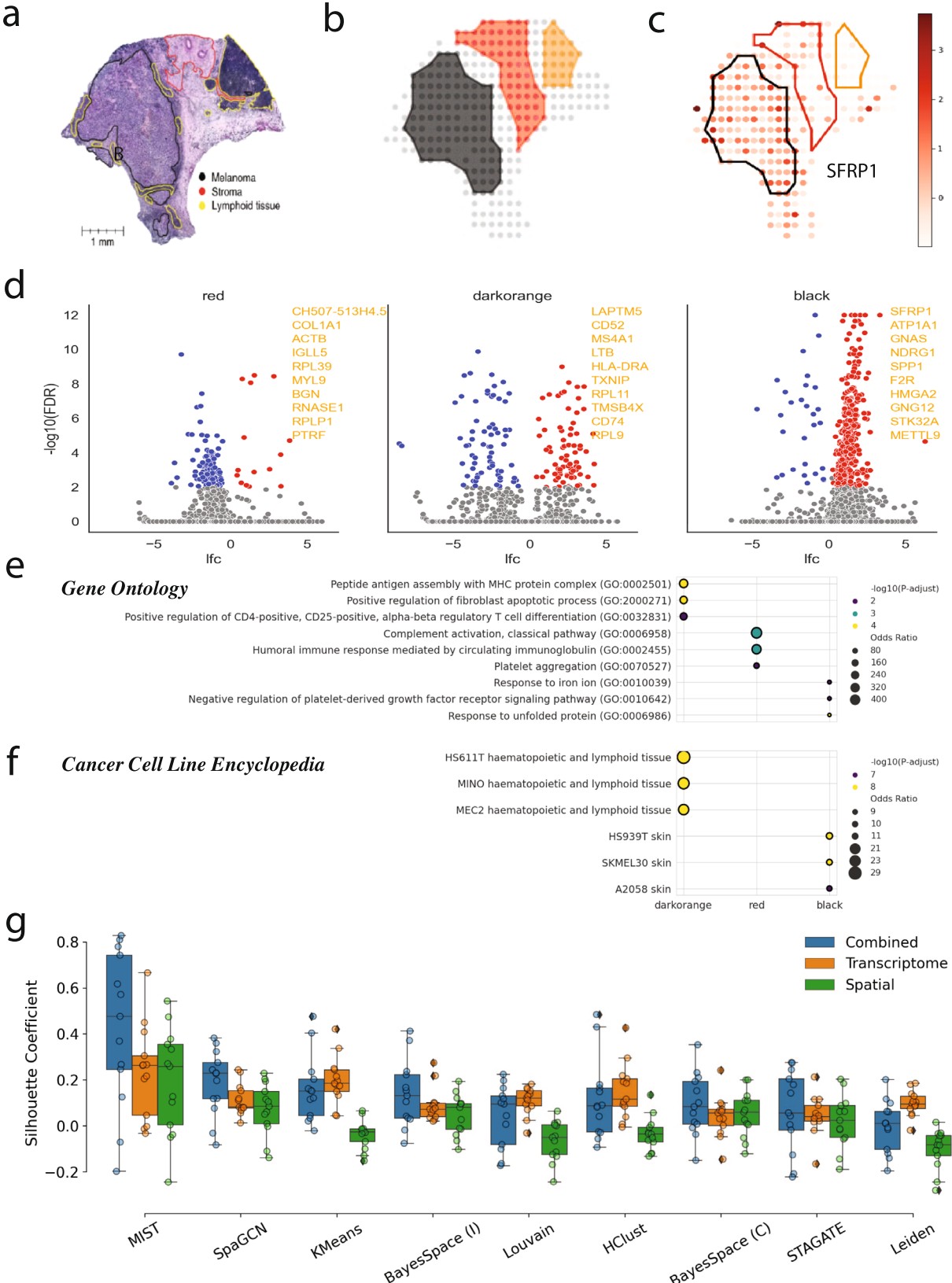

immunoglobulin" was significantly enriched in the red region (FDR = $3 \times 10^{-4}$, hypergeometric test adjusted by multiple comparisons), consistent with it being a stromal region (Fig. 2e).

To determine if any of the detected molecular regions is related to cancer pathology, we performed additional GSEA using gene sets from the Cancer Cell Line Encyclopedia[19]. Indeed, the black (tumor) region

was enriched in skin cancer cell lines, while the dark orange (lymphoid) region was enriched of lymphoid cell lines (Fig. 2f). We obtained similar accuracy in detecting molecular regions and providing functional information on pathology when we applied MIST to nine other tumor samples of various cancer types and tissues (Supp. Fig. 3–13, Supp. Data 1). MIST is thus generalizable as it accurately detects molecular

**Fig. 2 | MIST-detected molecular regions agree with manual annotation of tumor regions based on H&E. a** Manual annotation of a melanoma tumor sample on a pathologically stained image. **b** Three molecular regions detected by MIST: red, black and dark orange. **c** Localization of the SFRP1 expression heatmap with region boundaries. Increasing intensity of red indicates higher expression values. **d** Volcano plot of significantly enriched (red dots) and repressed (blue dots) genes in each detected region compared against spots from all other regions, with the top 10 significantly activated genes listed on the margin. Panels from left to right represent different regions: red (stroma), dark orange (lymphoid) and black (tumor). **e**, **f** Dot plots show Gene Set Enrichment Analysis (GSEA) results using Gene Ontology (**e**) and Cancer Cell Line Encyclopedia (**f**). *X* axis indicates regions and *y* axis represents each gene ontology term (**e**) or cancer-related cell line terms (**f**). Dot sizes represent odds ratios and colors represent statistical significances that

increase from blue to yellow. The statistical significances were derived from hypergeometric tests and adjusted for multiple comparisons. For each region, top three significant terms are shown. **g** Box plots of Silhouette coefficient scores using MIST and other clustering methods using transcriptomic profile (orange), spatial coordinates (green) and a combination of both (blue) ($n = 33473$ spots from 13 independent ST samples, Supp. Table 1). *X* axis (left to right sorted by Combined Silhouette coefficient): MIST, SpaGCN, K-means, BayesSpace with instructed parameters (I), Louvain clustering, hierarchical clustering (HClust), BayesSpace with the same number of regions (C) as MIST, STAGATE and Leiden clustering. Boxplots are defined with center line (median), box limits (upper and lower quartiles) and whiskers that extend at most 1.5 times of the interquartile range. Source data are provided as a Source Data file.

regions in different ST samples and provides clues to facilitate their functional annotations using molecular differences and GSEA.

## MIST-detected regions outperform all other methods in the preservation of the molecular and spatial structure of tissues

As expert-curated spot-level annotation is rarely available, evaluation based on spot-level ground truth is infeasible. Yet, it is essential to evaluate the accuracy of MIST's molecular region assignments at both the transcriptomic and spatial levels. To achieve this, we calculated the Silhouette coefficients of MIST's region assignments on the molecular profiles (referred to as the Transcriptome-level Silhouette Coefficient, TSC) and the spatial coordinates (referred to as the Spatial-level Silhouette Coefficient, SSC). Additionally, we summed these two scores as a combined Silhouette Coefficient (CSC) for each dataset (see Methods). We benchmarked MIST's TSC, SCC, and CSC with other spot-level clustering methods, including BayesSpace[12], STAGATE[20], SpaGCN[21], K-Means[22], Leiden[23], Louvain[23–25], and Hierarchical clustering[26], using twelve publicly available datasets that contained number of spots ranging from a few hundred to several thousand (Fig. 2g, Supp. Table 1).

A good model is expected to have high scores in all of these three metrics. In general, MIST ranked the highest for all measures amongst all compared methods (median TSC 0.26, SSC 0.26, and CSC 0.48) (Supp. Table 2-4). At the transcriptomic level, MIST outperformed other methods with a median improvement of 25% over the second-best performer, K-Means. At the spatial level, MIST outperformed other methods with a median improvement of 51% over the second-best performer, BayesSpace. When combining both scores, MIST outperformed all other methods across the tested datasets ($p \leq 0.01$, two-sided paired *t* test). We further demonstrated MIST's region detection accuracy using adjusted rand index scores on 12 Human Dorsal Frontal Cortex datasets (Supp. Fig. 27, Supp. Tables 5–6).

## MIST accurately recovers holdout values across multiple datasets

To assess MIST's accuracy when estimating missing values, we performed five-fold random holdout experiments in which we withheld a random set of the observed non-zero values and used these as a "ground truth" to evaluate the performance of several models. By withholding some of the observed values, we simulated cases in which non-zero expression values have dropped-out.

To consider the heterogeneity of datasets that might lead to biased performance, we tested 13 datasets (Supp. Table 1). These samples vary widely in the number of genes, number of spots, and sparsity levels. For each dataset, we selected genes that are expressed across at least half of all spots in the sample to generate holdout test datasets. For each gene, we partitioned the non-zero expression values into five non-overlapping sets. Then, we iteratively held-out one-fold of the values and assessed the accuracy of several methods in recovering the held-out gene expression values: we benchmarked MIST against MAGIC[27], knn-smoothing[28], McImpute[13], SAVER[29], DeepImpute[30], and a

baseline k-nearest neighbor method we constructed (spKNN) that estimates missing values by averaging spatially adjacent neighbors. To evaluate the accuracy of missing-value estimation in holdout experiments, we used Rooted Mean Square Error (RMSE) and Pearson Correlation Coefficients (PCC), where RMSE represents the error and PCC shows the agreement between the ground truth and estimated values. Better imputation methods should have lower RMSE and higher PCC scores.

MIST consistently outperformed the other methods across all datasets, with higher PCC and lower RMSE scores during holdout value evaluation (Fig. 3a, b). MIST had an average RMSE improvement (lower value) of 13% ($p = 8 \times 10^{-26}$, two-sided paired *t* test) compared with McImpute, 26% ($p = 10^{-54}$, two-sided paired *t* test) compared with MAGIC, and 61% ($p = 10^{-35}$, two-sided paired *t* test) compared with the baseline spKNN algorithm. MIST's mean PCC was also 8% larger than McImpute ($p = 3 \times 10^{-30}$, two-sided paired *t* test), 8% greater than MAGIC ($p = 2 \times 10^{-30}$, two-sided paired *t* test) and 55% greater than spKNN ($p = 1 \times 10^{-37}$, two-sided paired *t* test). Knn-smoothing and SAVER consistently performed substantially worse than the other methods (Supp. Figs. 14–15).

To investigate how gene sparsity affects denoising, we stratified the performance assessment at a per-gene-level, grouped by the sparsity level (zero-value proportion). While MAGIC and spKNN's estimation error monotonically increased with sparsity level, MIST's performance was not affected by gene sparsity (Fig. 3c). McImpute's performance was also not influenced by gene sparsity, but MIST outperformed McImpute at every gene sparsity level (Fig. 3c). MIST also faithfully recovered the gene expression spatial pattern for GAPDH in the melanoma tissue sample after denoising (Fig. 3d–f). With the holdout input, MIST accurately estimated the original expression values by increasing Spearman's correlation coefficient from 0.65 to 0.96 (Fig. 3g, h). When evaluating all genes across the tested datasets, after denoising, the median correlation rose from 0.64 to 0.88 (Fig 3i, Supp. Figs. 16–17).

## MIST discovered intra-cortical heterogeneity in an Alzheimer's Disease mouse model

To determine whether MIST could improve the clustering results of ST data, we turned to highly complex mouse brain. We used published ST data[7] and applied Uniform Manifold Approximation and Projection (UMAP)[31] to reduce the dimensionality of mouse brain gene expression data and visualize the clustering structures. First, we performed UMAP on a wild-type C57BL/6 J mouse brain sample with the raw and denoised transcriptomes, respectively (Fig. 4a). After denoising, MIST enhanced the heterogeneity of the data compared to the raw transcriptomes, with most of the spots within the cortex, hippocampus, and thalamus forming individual clusters (Fig. 4a, Supp. Fig. 18).

Similarly enhanced clustering patterns were identified in a brain sample from a App^{NL−G−F} mouse model of Alzheimer's disease (AD) after denoising. Here, however, the AD cortex separated into two

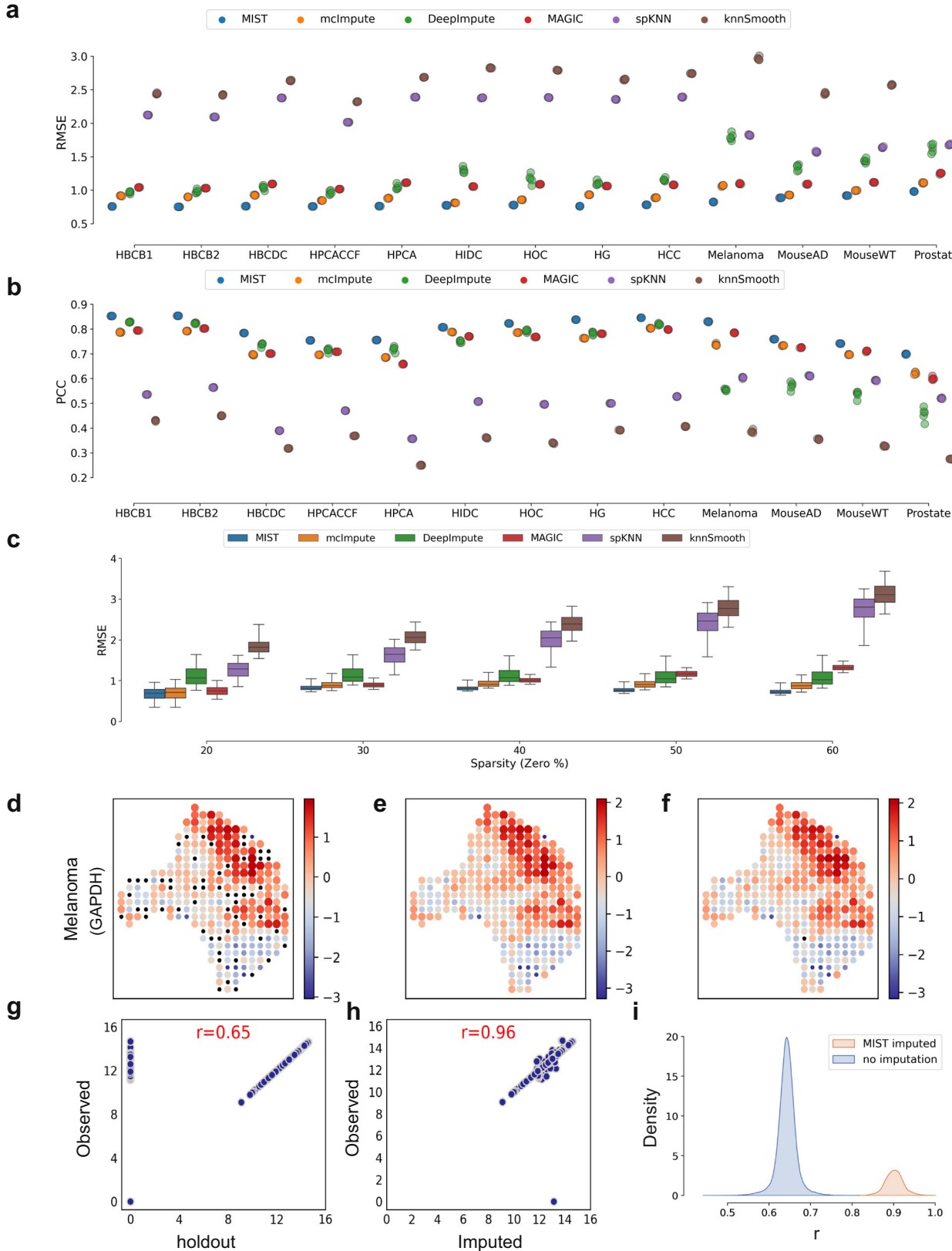

individual clusters, something we did not observe in either the wild-type mouse brain or in the original mouse AD data (Fig. 4b, Supp. Fig. 19). Further analysis revealed a clear separation into two parts (Fig. 4c, Supp. Fig. 20): cluster 1 consisted of the cortical subplate, olfactory, entorhinal, ectorhinal, temporal association, and perirhinal areas (as designated in the original paper), while cluster 2 contained the auditory, primary somatosensory, posterior parietal association, and retrosplenial areas. When we mapped these two clusters to the anatomical reference, cluster 1 occupied the upper quadrant while cluster 2 occupied the lower quadrant (Fig. 4d–f). This heterogeneity was detected only in the MIST-denoised AD cortex, and not the wild-type mouse cortex.

**Fig. 3 | MIST outperforms other imputation methods in holdout experiments.**
**a–b** Dot plots of (**a**) rooted mean square error (RMSE) and (**b**) Pearson's correlation coefficient (PCC) in the fivefold holdout experiment using multiple datasets. Each column on the *X* axis is an individual dataset (Supp. Table 1). Colors of dots represent different benchmarked imputation methods as shown in the legends. **c** Boxplot of gene-level performance of each model represented by the RMSE (*y* axis). *X* axis groups genes by sparsity level (zero-value percentage) of 20% (*n* = 2078), 30% (*n* = 4061), 40% (*n* = 5761), 50% (*n* = 8077) and 60% (*n* = 6078). Colors of boxes represent different benchmarked imputation methods as shown in the legend. Boxplots are defined with center line (median), box limits (upper and lower quartiles) and whiskers that extend at most 1.5 times of the interquartile range. **d–f** Expression patterns recovered for gene GAPDH in the melanoma sample. From left to right: the spatial pattern of the input to MIST with holdouts denoted by black dots (**d**), the denoised gene expression pattern (**e**), and the original observed gene expression pattern (**f**). Color and size indicate relative gene expression abundance. **g** The Spearman's rank correlation coefficient (*r*) between holdout input and observed GAPDH expression values. **h** The correlation between MIST-denoised and observed GAPDH expression values. **i** Distribution of the gene-level correlation between original non-zero gene expression values and the holdout input (blue), and denoised values (orange), respectively. Source data are provided in the Source Data file.

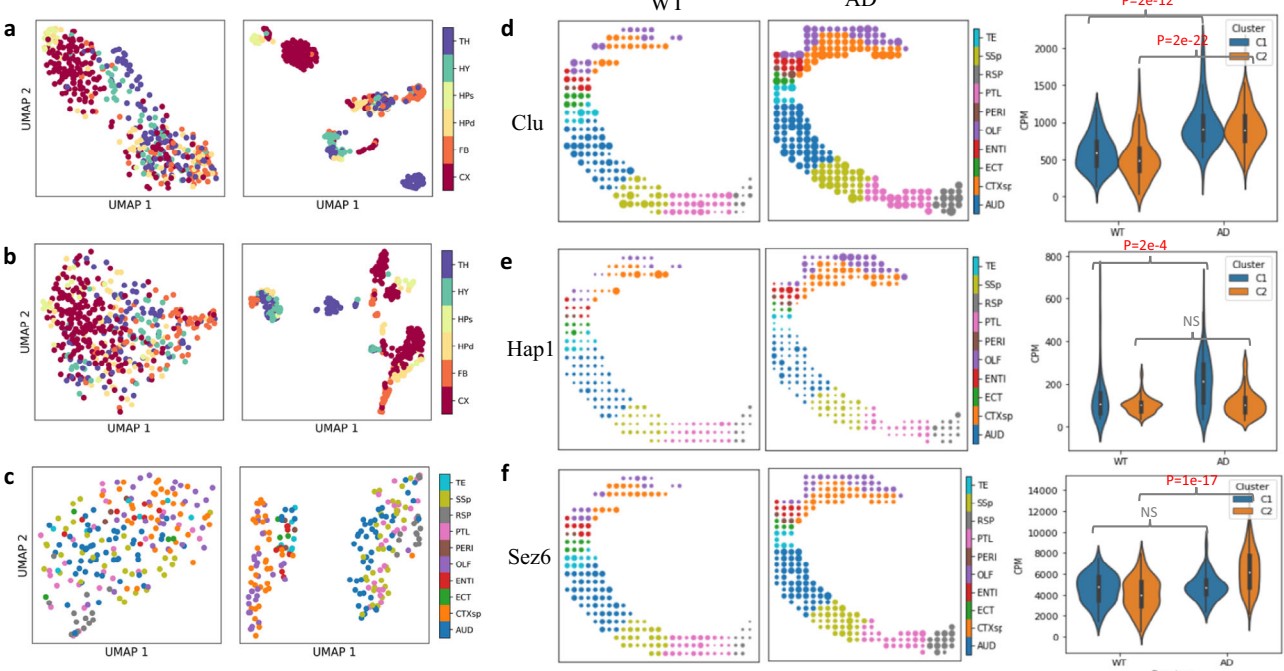

**Fig. 4 | MIST-identified intra-cortex heterogeneity within Alzheimer's Disease (AD) mouse brain. a** UMAP of the mouse wild-type (WT) brain sample[7] using original observed ST data (left) and MIST-denoised ST data (right) colored by region names (TH: thalamus, HY: hypothalamus, FB: fiber tract, HPd: dendritic layer of the hippocampus, HPs: somatic layer of the hippocampus, CX: cortex). **b** UMAP of the mouse AD brain sample[7] using original observed ST data (left) and MIST-denoised ST data (right) colored by region names. **c** UMAP of the mouse AD cortex using original observed ST data (left) and MIST-denoised ST data (right) colored by sub-cortex region names (CTXsp: cortical subplate, OLF: olfactory, ENTI: entorhinal, TE: temporal association, ECT: ectorhinal, and PERI: perirhinal areas, AUD: auditory, PTL: posterior parietal association area, RSP: retrosplenial area, SSp: primary somatosensory). **d–f** Examples of the AD upregulated genes that are activated in both clusters (**d**, WT: *n* = 172 spots, AD: *n* = 209 spots), Cluster 1 only (**e**, WT: *n* = 61 spots, AD: *n* = 87 spots) and Cluster 2 only (**f**, WT: *n* = 111 spots, AD: *n* = 121 spots). Left: spatial expression pattern in the WT mouse cortex (colors represent regions and sizes represent expression levels); middle: spatial expression pattern in the AD mouse cortex; right: violin plots containing box plots of gene expression levels grouped by genotype (WT/AD) and spatial clusters (C1 in blue and C2 in orange). Boxplots are defined with center line (median), box limits (upper and lower quartiles) and whiskers that extend at most 1.5 times of the interquartile range. The statistical significances were derived from two-sided Wilcoxon rank-sum tests without multi-comparison adjustments. Significant results were shown with a p-value in red while insignificant results are labeled as NS in gray. Source data are provided in the Source Data file.

To then understand the biological importance of these two clusters in AD pathology, we performed differential gene analysis to extract AD-activated genes from each. We selected upregulated genes in the AD sample with a fold change >50% and adjusted *p* < 0.01 (two-sided Wilcoxon rank-sum test), resulting in 55 markers for cluster 1 and 41 markers for cluster 2 (Supp. Figs. 21–22). Only 21 AD-activated genes, such as Clu[32], were shared between these two clusters (Fig. 4d). Thirty-four genes, including Hap1[33], were upregulated only in cluster 1 (Fig. 4e) and 20 genes, such as Sez6[34], were upregulated only in cluster 2 (Fig. 4f). The spatial clusters in MIST-denoised cortical data reveal heterogeneity in the transcriptional response to AD. For example, the entorhinal cortex, known to be among the first regions to become dysfunctional in AD[35], showed significant upregulation of Hap1 (Fig. 4e) compared to other cortical regions. This divergence can be seen in the clusters (Fig. 4c, right) as well as in the spatial data (Fig. 4e). Similarly,

Sez6 was upregulated only in the auditory cortex, whose functionality is impaired in AD[36] (Fig. 4f). These results demonstrate that MIST can extract biological insights from ST data.

## MIST recovers spatial gene-gene co-expression patterns

Dropouts within ST datasets weaken the correlation analysis and cause inaccurate estimation of gene-gene spatial correlation, which is the fundamental element in many analyses such as weighted correlation network analysis (WGCNA)[37]. To test MIST's ability to recover spatial co-expression patterns, we examined two pairs of genes: Cldn11-Arhgef10 and Gfap-Aqp4. Based on the Human Protein Atlas[38], CLDN11 and ARHGEF10 have enhanced protein expression levels in oligodendrocytes while GFAP and AQP4 are enhanced in astrocytes.

Cldn11-Arhgef10 showed a high spatial correlation score of 0.97 based on the reference Allen Brain Atlas[39] (Fig. 5a), but the original ST

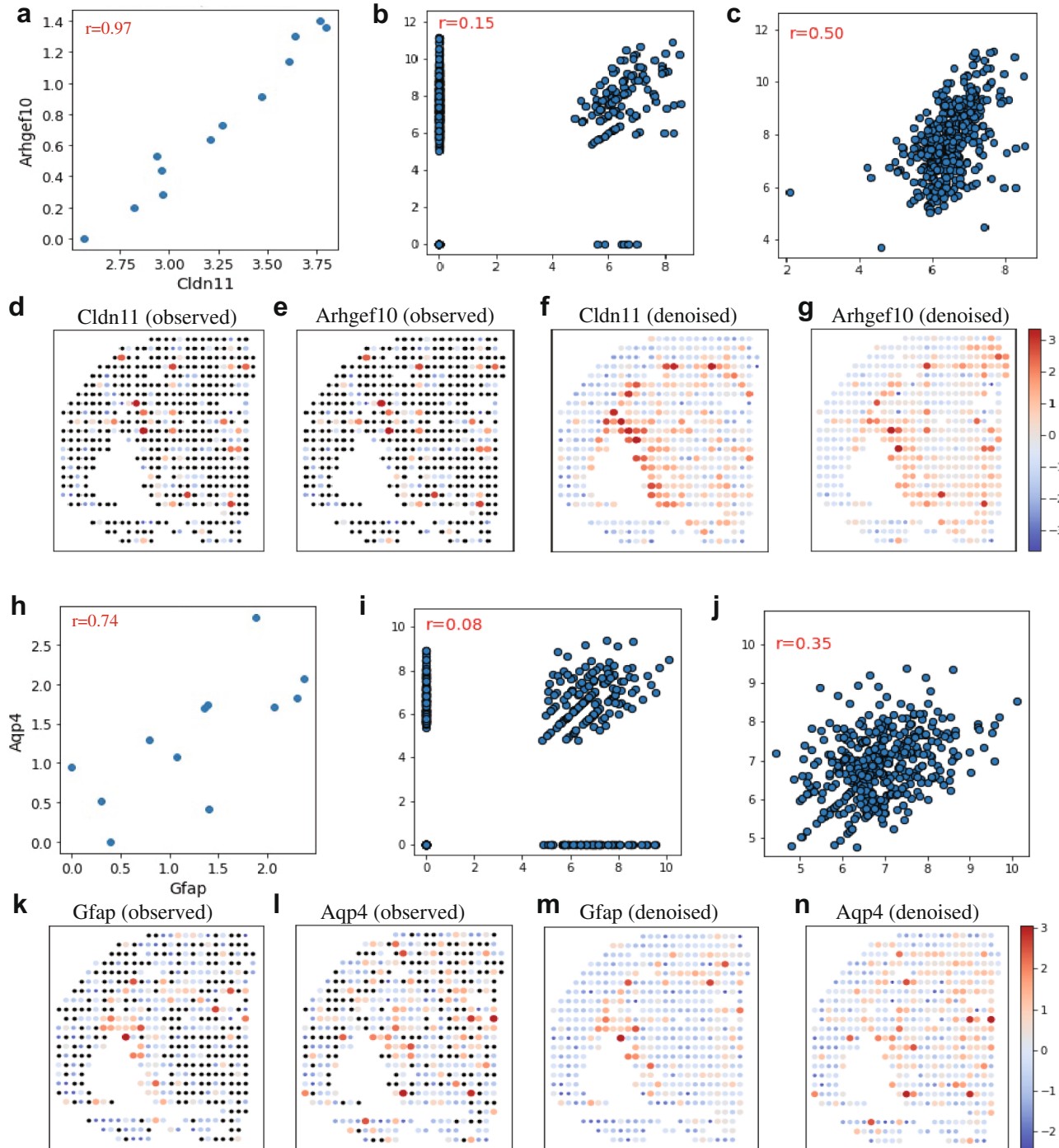

**Fig. 5 | MIST recovers spatially co-expressed gene pairs. a–c** Spatial correlation of Arhgef10-Cldn11 in the Allen Brain Atlas reference (**a**), the original ST data (**b**), and the MIST-denoised ST data (**c**). **d** Expression patterns of the Cldn11 using the original ST data; zero values are colored black and non-zero values are z score scaled for visualization. **e** Expression pattern of the Arhgef10 using the original ST data. **f** Expression pattern of the Cldn11 using the MIST-denoised data. **g** Expression patterns of the Arhgef10 using the MIST-denoised data. **h–j** Spatial correlation of Gfap-Aqp4 in the Allen Brain Atlas reference (**h**), the original ST data (**i**), and the MIST-denoised ST data (**j**). **k** Expression patterns of Gfap using the original ST data. **l** Expression patterns of Apq4 using the original ST data. **m** Expression patterns of Gfap using the MIST-denoised data. **n** Expression pattern of Aqp4 using the MIST-denoised data. Source data are provided in the Source Data file.

data yielded a correlation score of only 0.15 (Fig. 5b). Another single-cell study also confirmed their high co-expression at both single-cell and pseudo-bulk levels (Supp. Note 1, Supp. Fig. 23–25). After denoising by MIST, the correlation score was improved to 0.5 (Fig. 5c). To visualize the gene expression patterns, we plotted the heatmap of log-scale expression values and showed that Cldn11 and Arhgef10 have similar gene expression patterns only after MIST denoising (Fig. 5d–g).

To test whether MIST can recover co-expressed gene pairs that are not significantly correlated with the original data, we examined the second pair of genes, Gfap-Aqp4, which have a moderately good spatial correlation with a score of 0.74 in the reference Allen Brain Atlas database (Fig. 5h), also positively correlated in an external single-cell mouse brain cohort (Supp. Note 1, Supp. Fig. 23–25). Before denoising, we observed insignificant correlation with a score of 0.08 ($p = 0.56$,

Spearman's correlation test, Fig. 5i). After denoising, we recovered a significant spatial correlation with a score of 0.35 ($p = 9 \times 10^{-12}$, Spearman's correlation test, Fig. 5j). Similar to the first pair of genes, the co-expression of Gfap-Aqp4 could be observed only after denoising by MIST (Fig. 5k−n).

These two cases demonstrate that MIST restores the spatial correlation of gene pairs whose co-expression patterns are either lessened or lost in the original ST data. Given that co-expression estimation is vital in many downstream analyses such as identifying gene modules[37], MIST's ability to denoise ST data before carrying out such analyses provides a substantial improvement over current methods.

## Discussion

In this study, we developed an algorithm for processing of spatial transcriptomics data, MIST. We show that MIST overcomes two major problems encountered in ST data analyses: in silico region detection and missing-value estimation. MIST solves the first challenge by combining molecular similarity and spatial connectivity between spots and enabling automated, unbiased region detection. After region detection, MIST facilitates users to annotate each region by comparing regional activated genes to established gene sets through GSEA. Unlike other clustering methods that assign a membership to every spot, MIST leaves some spots that are likely to be regional boundaries unannotated. We view these isolated spots as important elements for domain-specific studies, such as tumor microenvironments.

MIST solves the second challenge by a mini-batch and region-based low-rank approximation. This is based on a simple yet interpretable assumption that the number of cell types for any given region is small. This assumption has been adopted by many other single-cell RNA-sequencing denoising methods such as McImpute[13] and ALRA[40]. Compared with McImpute and ALRA, however, MIST significantly improved denoising specificity using region-specific imputation. We further demonstrated that MIST recovers spatial patterns of co-expressed genes that are highly correlated in a reference atlas but poorly correlated in the original ST data. Since many downstream analyses such as WGCNA[37] are based on co-expression estimation, denoising ST by MIST would avoid false conclusions due to inaccurate co-expression estimation.

MIST was designed to address dropout issues in the 10X Visium platform and reduce the noise on measured genes. 10X does provide a software called Space Ranger that can process the raw sequencing files and the aligned histological images, but it does not provide functions for imputation or region boundary detection. There are also some other spatial methods implemented in different languages such as ScanPy[10] (in Python) and Seurat[11] (in R). These methods allow users to process the data, perform clustering as in the traditional scRNA-seq pipeline, and then visualize the cluster behaviors, but they can neither denoise the data through imputation nor integrate spatial information into the clustering, which they base solely on molecular similarity. Instead, MIST allows the user to identify regions that are both molecularly similar and spatially adjacent. MIST's abilities are thus *sui generis* in the ST field and will enable analysis of ST data with higher accuracy.

MIST will enable researchers to recover important biological signals in downstream analyses, such as when identifying spatial gene-gene co-expression patterns. While the original ST data provided by 10X Visium might suffer from the sparsity issue, MIST accurately recovers the missing values, which increased the expression signals and revealed the genes' spatial co-expression patterns. We envision that MIST will be useful for identifying local subregions within tumors and finding unexpected patterns of spatial organization in tissues undergoing developmental or pathological changes.

Despite the functionalities enabled by MIST, we see its limitation. Firstly, since MIST detects regions with high intra-region similarity and inter-region dissimilarity, some spots will be left-out. To mitigate this issue, we allowed users to tune the number of left-out spots by imposing a penalty parameter. Although the current MIST software did not

comprehensively analyze these left-out spots, we will develop further analysis on them to help understand the communications between different regions in future work. Another limitation is that MIST does not allow cross-sample analyses in one run, which might improve the analyses by mutual referencing. Although the current version of MIST is performed at a slide-by-slide level, enabling integrative analyses is of great interest in the future improvement of the algorithm.

## Methods

### Data collection and preprocessing

In this study, we included twelve spatial transcriptomics datasets that varied in their number of spots, number of genes, and sparsity levels (Supplementary Table 1). Every dataset has a raw mRNA count matrix form where rows indicate spots and columns indicate genes.

To filter out low-quality genes that might otherwise introduce noise to the pipeline, we kept genes that are observed in at least ten spots. To remove low-quality spots, we filtered out spots with total UMI counts <1500 in the tissue or with > 30% mitochondrial genes.

To account for the different library sizes of every spot due to variance in sequencing depth and the number of cells, we normalized the raw mRNA count matrix using count per million (CPM = $\frac{\text{Raw count} \times 10^6}{\text{Library size}}$). Data preprocessing and normalization procedures were conducted using the Python package ScanPy[10].

### Spatial graph construction

Suppose the ST expression matrix has $M$ spots and $N$ genes, the spatial gene-expression profile can be defined as $\mathbf{Y} \in \mathbf{R}_*^{M \times N}$, where $\mathbf{Y}$ is the observed gene expression matrix, and $\mathbf{R}_*$ denotes non-negative real matrices with $M$ rows and $N$ columns. The $M$ spots in an ST slide can form a lattice graph, $\mathbf{G} = <\mathbf{V}, \mathbf{E}>$, where $\mathbf{V}$ is the node-set and $\mathbf{E}$ is the edge-set. Every pair of adjacent $(u, v)$ spots are connected with edge $\mathbf{E}$ $(u, v)$.

### Weight calculation

To infer the weights for every connected edge $\mathbf{E}$ $(u, v)$, we calculated the Pearson correlation coefficient between the gene expression profile of spot $u$ and $v$. To remove the noise in high-dimensional gene expression data while keeping the major signals, we extracted the top 80% highly variable genes and used Principal Component Analysis (PCA)[41] to reduce the dimensions of the gene expression matrix. We kept the first $p$ principal components with a default $p$ of 30. The weight for edge $\mathbf{E}$ $(u, v)$ was then inferred using the correlation score between the first $p$ principal components of spot $u$ and $v$ (Supp. Algorithm 1).

### Edge pruning and region detection

To draw the boundaries between functionally dissimilar regions, we removed edges whose weights are lower than a threshold $\varepsilon$.

To detect the regions within tissues, we used a depth-first search algorithm[42] to identify all the connected components in graph $\mathbf{G}$ after edge removal (Supp. Algorithm 1). Every connected component with more than $q$ spots are identified as a region. In default, a $q$ of 40 is used for Visium ST data and a $q$ of 20 is used for the other ST samples with <500 spots in the tissue sample.

### Parameter selection by mathematical optimization

To automatically select the threshold and avoid bias, we did a grid search with positive threshold values ranging from 0.1 to 0.9 (Supp. Algorithm 2). Specifically, we optimized the threshold value using the following equation:

$$Max. \frac{\sum_{\mathbf{r} \in \mathbf{R}} \frac{\sum_{i \in \mathbf{r}, j \in \mathbf{r}, i \neq j} \text{sim}(i,j)}{|\{i,j\}|i \in \mathbf{r}, j \in \mathbf{r}, i \neq j|}}{|\mathbf{R}|} - \frac{\sum_{\mathbf{r} \in \mathbf{R}, \mathbf{r}' \in \mathbf{R}, \mathbf{r} \neq \mathbf{r}'} \frac{\sum_{i' \in \mathbf{r}, j' \in \mathbf{r}'} \text{sim}(i',j')}{|\{i',j'\}, i' \in \mathbf{r}, j' \in \mathbf{r}'|}}{|\{\mathbf{r}, \mathbf{r}'\}, \mathbf{r} \in \mathbf{R}, \mathbf{r}' \in \mathbf{R}, \mathbf{r} \neq \mathbf{r}'|} + \sigma^* \frac{\sum_{\mathbf{r} \in \mathbf{R}} |\mathbf{r}|}{M}$$

$$(1)$$

In Eq. 1, the first term maximizes the average intra-region similarity with sim($i, j$) defined as the correlation of spot $i$ with spot $j$ within region **r**. **R** represents the set of all regions and |**R**| is the number of regions detected. The second term minimizes the average inter-region similarity. The third term maximizes the coverage of detected regions and minimizes the spots that are defined as isolated spots, which are typically boundary spots that have a mixture of cells from multiple regions. The third term is regulated by a hyper-parameter $\sigma$ with a default value of 0.1.

## Low-rank matrix completion

Suppose the observed gene expression matrix for region **r** is $\mathbf{Y_r}$, where $\mathbf{Y_r}$ is a sparse matrix with $M$ rows (spots) and $N$ columns (genes). The task is to estimate $\mathbf{X_r}$, which represents the denoised gene expression matrix for region **r**. We adapted the low-rank-matrix completion algorithm through singular value decomposition used by McImpute[13].

Given the assumption that the number of cell types within a functional region is small, we expect $\mathbf{X_r}$ to have a low-rank. To achieve this goal, the task is turned into a low-rank matrix completion problem by solving the following objective function:

$$\min_{\mathbf{X_r}} ||\mathbf{Y_r} - A(\mathbf{X_r})||^2 + \lambda^* \text{rank}(\mathbf{X_r}) \tag{2}$$

The first term in Eq. (2) minimizes the error between the non-missing gene expression values of $\mathbf{X_r}$ and $\mathbf{Y_r}$ with a projection function A that returns values in $\mathbf{X_r}$ at the indices of non-missing values in $\mathbf{Y_r}$. The second term in Eq. (2) minimizes the rank of the denoised gene expression matrix. The objective function is a linear combination of these two terms regularized by a non-zero tuning parameter $\lambda$. Theoretically, a larger $\lambda$ will give us a lower-ranked denoised gene expression matrix whose values on the non-missing indices might deviate from the ground truth. On the other hand, a small $\lambda$ will result in a relatively high-rank denoised matrix with a lower error on the non-missing indices.

However, minimizing the rank of a matrix is non-convex. To transform it to a convex problem with a globally optimal solution, we relaxed Eq. 2 as:

$$\min_{\mathbf{X_r}} ||\mathbf{Y_r} - A(\mathbf{X_r})||^2 + \lambda^* ||\mathbf{X_r}||_{\text{nuc}} \tag{3}$$

where we transformed the term 2 in Eq. (2) as a nuclear norm of $\mathbf{X_C}$, which can be calculated by summing up the singular values obtained through singular value decomposition. Specifically, Eq. (3) can be further transformed as

$$\min_{\mathbf{X_r}} ||\mathbf{B} - \mathbf{X_r}||^2 + \lambda^* ||\mathbf{X_r}||_{\text{nuc}} \tag{4}$$

where $\mathbf{B_{k+1}} = \mathbf{X_{k,r}} + \frac{1}{\alpha} A^T(\mathbf{Y_r} - A(\mathbf{X_{k,r}}))$. Using the inequality $||\mathbf{B} - \mathbf{X}||_2 \geq ||s_\mathbf{B} - s_\mathbf{X}||$, where $s_\mathbf{X}$ denotes the singular value vector for matrix $\mathbf{X}$, Eq. (4) can be rewritten as:

$$\min_{X_C} ||s_\mathbf{B} - s_{\mathbf{X_r}}||^2 + \lambda^* ||s_{\mathbf{X_r}}||_1 \tag{5}$$

Therefore, by taking the derivative of Eq. (5), $s_{\mathbf{X_r}}$ is solved by soft thresholding the singular values of $s_\mathbf{B}$ with a threshold equal to $\lambda/2$.

To tune the parameter $\lambda$ to strike the balance between low matrix singularity and low error on non-missing indices, we find the maximal $\lambda$ that achieves a fixed low error ($10^{-12}$) calculated by the sum of the absolute difference between denoised and observed values on non-missing indices.

## Imputation using region-based mini-batch matrix completion

Ensemble methods are used to boost prediction performance by aggregating the results from multiple weak learners. Random Forest classifiers are a classic example. In random forest, a set of decision trees are trained with sampled features and samples[43]. The final prediction is made by averaging the prediction results from the decision trees.

Recently, because of the greater resolution of biological data, e.g., thousands to millions of single cells, mini-batch ensemble-based machine learning models are proposed to cluster the samples to achieve accuracy while reducing the computational time. For example, mbkmeans[44] randomly and repeatedly samples single cells from large-scale single-cell data, without placement, to perform K-Means clustering. However, such methods can lack important biological information because the samples selected into each batch are random.

To leverage mini-batch-based ensembles to achieve more accurate but not computational-expensive imputation while preserving biological information, we used a region-based mini-batch matrix completion methods for imputation (Supp. Algorithm 3). In brief, for each region **r**, $k$ mini-batches are sampled with each batch containing all spots in **r** with random spots selected equally from two resources: isolated spots and other core regions. After running region-based imputation (Supp. Algorithm 1 and 2) $k$ times, the final imputed value is estimated by averaging the results from the mini-batches. Each isolated spot will also be imputed by averaging results with size of |**R**| from all regions. We showed that mini-batch-based approach improved the performance in the holdout experiments comparing against without mini-batches (Supp. Fig. 26).

## Evaluating region detection accuracy in benchmarking

Due to the lack of spot-level annotations for all the samples except the mouse brain samples (Supp. Table 1), we used an internal evaluation method (Silhouette Coefficient) to evaluate the performance of region detection and compare it with other methods.

We calculated two Silhouette Coefficients, one transcriptome-based and one based on spatial coordinates, using silhouette_score(**a**, labels) function from module "metrics" of Python package scikit-learn[45], where **a** is the top $p$ principal components for the transcriptome-level Silhouette coefficient (TSC) evaluation, and $(x, y)$ coordinates for the SSC evaluation. Intuitively, either TSC or SSC ranged from negative one to positive one, with negative values representing wrong clustering and a positive value meaning accurate clustering results.

## Running other clustering methods

We ran BayesSpace, SpaGCN, Leiden, Louvain, and K-Means clustering to compare them with MIST in the accuracy of their region detection abilities. We specified the number of clusters in BayesSpace, SpaGCN, and K-Means to be the same as MIST's detected molecular regions. For Leiden and Louvain, which don't require such parameters, we used their default settings.

To make a fair comparison, we ran BayesSpace in two modes. The instructed mode uses the parameters with the top 2000 highly variable genes and 15 principal components instructed by the authors (https://edward130603.github.io/BayesSpace/articles/maynard_DLPFC.html), and the comparable mode uses the same parameters as MIST with top 80% highly variable genes and 30 principal components. To run BayesSpace, we used the spatialCluster() function from BayesSpace R package[12].

We ran SpaGCN[21] by following the tutorial (https://github.com/jianhuupenn/SpaGCN/blob/master/tutorial/tutorial.ipynb) provided by the authors.

To run STAGATE[20], we followed the tutorial (https://stagate.readthedocs.io/en/latest/T1_DLPFC.html) provided by the authors.

To run Leiden and Louvain clustering, we used the Python package ScanPy[10] with the top 80% highly variable genes and 30 principal components, and other parameters as default.

To run K-Means and hierarchical clustering (HClust), we used the KMeans() and AgglomerativeClustering() functions from cluster module of the sklearn[45] Python package with the input as the reduced feature space (top 30 principal components).

### Differential gene expression analysis

Wilcoxon rank-sum test provided by the Python package Scipy[46] was used to infer the significant level of differentially expressed genes. Fold change of genes between condition and control was calculated based on the difference of average gene expression within groups. To get regional differentially expressed genes, we compared the spots within the target region against spots from all other regions. Then, we selected genes with a fold change >50% and adjusted $p$ value <0.01.

### Gene set enrichment analysis

The Python package GSEApy was used to perform gene ontology enrichment analysis using the list of differentially expressed genes for each region.

### Data generation for holdout experiments

To improve the diversity in the holdout experiments, we tested samples including a mouse wild-type brain sample, a mouse AD brain sample, a melanoma tumor sample, a prostate tumor sample, and nine other human tumor samples. To extract good-quality genes to simulate the dropout events, we removed genes that were expressed in <50% of the spots.

To generate the holdout data, we used a five-fold cross-validation schema for the non-zero values. Specially, we first randomly partitioned every gene's non-zero expression values into five groups. In each holdout, we created missing values by setting one group to zero and performed imputation based on the remaining values. The held-out values served as ground truth for evaluating the accuracy of the imputation algorithms.

### Evaluating holdout experiments' performance

To quantifying the accuracy in recovering the held-out values, we reported two metrics including RMSE and PCC, where RMSE measures the estimation error and PCC measures the linear correlation between the true expression values and the estimated values. RMSE is defined as

$$RMSE = \sqrt{\frac{\sum_{i=1}^{n}(X_i - Y_i)^2}{n}}$$

and PCC is defined as

$$PCC = \frac{\sum_{i=1}^{n}(X_i - \bar{X})(Y_i - \bar{Y})}{\sqrt{\sum_{i=1}^{n}(X_i - \bar{X})^2}\sqrt{\sum_{i=1}^{n}(Y_i - \bar{Y})^2}},$$

where $Y$ is the holdout non-zero values, $X$ represents the MIST estimated values and $n$ denotes the number of holdout values.

To quantify the recovery of gene expression patterns after denoising the holdout data, we used Spearman's rank correlation test implemented by Scipy[46] to assess the correlation and the corresponding significance level between the original ST and denoised gene expression values.

### Co-expression analysis using the Allen Brain Atlas as a reference

We obtained mouse brain (coronal section) regional expression values for gene Cldn11 (experiment: RP_070116_01_G04), Arhgef10 (experiment: RP_070116_01_B05), Gfap (experiment: RP_Baylor_253913) and Aqp4 (experiment: RP_040324_01_F07) from Allen Brain Atlas[39] as references. Gene-expression values provided by Allen Brain Atlas are at the log2 scale.

Correlation scores between gene pairs in both reference and ST data are represented by Spearman's correlation coefficient calculated using the Python package Scipy[46].

### Reporting summary

Further information on research design is available in the Nature Portfolio Reporting Summary linked to this article.

## Data availability

The melanoma and prostate datasets were obtained from the link (http://www.spatialtranscriptomicsresearch.org/) provided in the original publications[4,5]. The mouse brain samples were obtained from the GEO database (accession number: GSE152506, sample N06_D2 and B06_E1, https://www.ncbi.nlm.nih.gov/geo/query/acc.cgi?acc=GSE152506)[7]. Other Visium datasets were downloaded from 10x Genomics (https://www.10xgenomics.com/resources/datasets). The Human DLPFC data was downloaded by following the instructions from spatialLIBD (http://spatial.libd.org/spatialLIBD/)[47]. Source data are provided with this paper.

## Code availability

The MIST algorithm is implemented in Python and is available at GitHub (https://github.com/linhuawang/MIST.git) and Zenodo (https://zenodo.org/badge/latestdoi/337148299)[48]. Code to reproduce the results for this manuscript is at https://zenodo.org/badge/latestdoi/486729457[49].

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

## Acknowledgements

Research reported in this publication was partially supported by the Eunice Kennedy Shriver National Institute of Child Health & Human Development of the National Institutes of Health under Award Number P50HD103555 for use of the Bioinformatics Core facilities. The content is solely the responsibility of the authors and does not necessarily represent the official views of the National Institutes of Health. Z.L. and L.W. are also partially supported by the Chao Endowment. We thank the handling editor and V. Brandt for editing this manuscript.

## Author contributions

L.W. and Z.L. conceived the project. L. W. developed and implemented the methods, collected the data, and performed the computational experiments and analyses. Z.L. supervised the project and led the discussions on the results. L.W. drafted the manuscript. M.M.S. provided critical input on the correlation of MIST-detected molecular regions to histological regions. All authors contributed to the final manuscript.

## Competing interests

The authors declare no competing interests.
