## [Peer Review File · Nature Communications]

Reviewers' Comments:

Reviewer #1:

Remarks to the Author:

Spatial transcriptomics developed in 2016 by Ståhl et al. (2016) and allows in situ capturing of transcriptome-wide gene expression on almost single-cell resolution, is now a mature technology. Similar to single-cell sequencing technologies some challenges remain in performing improved downstream analyses of ST data; these are mainly variation in gene expression caused by the efficiencies of the cDNA library formation and sequencing, technical dropouts, and batch effects. The application of normalisation techniques tries to address these problems. Another important part of ST data analysis is the identification of regions having a unique gene expression or gene expression profile.

The authors of the manuscript propose a method, called Missing-value imputation and in silico region detection for spatially resolved transcriptomics (MIST), to improve ST data quality by removing technical dropouts within regions of similar gene expression. The method first identifies regions with similar gene expression using a 2D graph after edge removal if such a region contains more than five spots, and second, it removes technical dropouts within such a region using iteratively low-rank matrix completions. The authors claim that MIST accurately recovers spatial transcriptomics missing values, identifies intra-tissue heterogeneity and recovers spatial gene-gene co-expression signals.

Major critic points

1) Manual Annotation

The authors criticise that tissue sections that were analysed with ST still require manual annotation by scientists, e. g. pathologists, to obtain knowledge about the cell type-specific structure within a tissue sample. However, even if applying the MIST method or any other clustering or dimension reduction method, the resulting regions have to be annotated by a scientist as these methods does not offer information, which cell type or gene expression profile was captured; only the spatial distribution of regions with similar gene expression is shown. To provide a method that indeed helps to annotate regions having similar gene expression or even obviates the work of an annotator, more information for each identified region needs to be provided.

2) Spatially connected regions of similar gene expression

The authors state that there is no method that addresses "the region-detection task in a way that establishes the boundaries of different tissue regions". Spatial Transcriptome Decomposition (STD) developed for ST data and used in Berglund et al (2018) to identify cancer, PIN3, inflammation, healthy epithelial, and stroma regions in multiple tissue sections and Spotlight (Elosua-Bayes et al., 2021) are at least two methods that were developed for ST data and address the identification of regions with a unique gene expression profile. Both also normalises ST data and removes technical dropouts. Another example of a clustering method especially developed for ST data is SpatialCPie (Bergenstråhle et al, 2020), which is applied to human heart tissue samples. It is recommended that the authors compare MIST with these methods.

3) Mixed cell types within tissue sections

The proposed method identifies spots having similar gene expression. However, if a tissue section contains mixed cell types besides regions of unique cell types, for example, a region of epithelial cells, a region of stromal cells, and a region of a mix of epithelial and stromal cells within one tissue sample, how does the proposed method address this?

4) Tissue examples

The authors apply the MIST method to three examples for which Spatial Transcriptomics was applied to. However, the mouse brain samples dominate the manuscript. To provide insights into the generalizability of the MIST method, it has to be applied to a broader number of ST tissue

samples. For example, the gene expression distribution of only one gene is shown in only one tissue sample of the prostate cancer example comprising twelve tissue samples and up to ~9000 transcripts per spot.

5) Multiple tissue samples

Although the authors do not claim that the MIST method can be applied to multiple tissue sections it can be a major part in analysing tissue sections. Then, batch effects can influence the analysis of multiple tissue samples. There is a risk when using MIST for multiple samples, that similar regions across tissue sections are missed. Can MIST applied to process multiple tissue sections? If not, this might be a major limitation of the proposed method.

6) Comparison to other methods

A major critical point is that the variance of gene expression due to technical problems in library formation and sequencing is neglected as well as batch effects if multiple tissue samples are analysed. Both problems are usually addressed using normalisation methods available for ST data and single-cell sequenced RNA (the latter can also be applied and is often used to analyse ST data). Therefore, the most common order in pre-processing ST data and preparing for downstream analysis is firstly normalisation including removal of batch effects, technical dropouts and variance in gene expression due to technical challenges (across multiple tissue samples) and secondly identification of regions (across similar tissue section) having similar gene expression or similar gene expression profiles.

Although normalisation across multiple tissue sample might remove biological meaningful differences.

To be able to compare the performance of the proposed method, the result should be compared with results for which both aforementioned steps (normalisation, clustering/dimension reduction) were applied, for example, using the regions identified in Berglund et al (2018) using STD. Further, it is recommended to use normalisation methods for a comparison that are very common for scRNA, for example BASiCS, SCnorm, and scran (Lytal et al., 202) although they consider not only technical dropouts.

7)

Just by eye, the overlap of the pathologist's annotation and the MIST regions shown in Figure 2d and 2f is not very convincing although higher than for the BayesSpace regions shown in Fig 2f.

8) Benchmarking

The authors use the adjusted Rand index (ARI) to benchmark the MIST method. Although ARI is an appropriate and common benchmark for clustering, it would be helpful to use additional benchmarks to compare the clustered ST data for which alternative methods were applied to, e.g. the Silhouette Coefficient.

Minor critic points

1)

What is the statistical motivation to use a log fold change cut off?

References

Lytal, N., Ran, D., & An, L. (2020). Normalization methods on single-cell RNA-seq data: an empirical survey. *Frontiers in genetics*, 11, 41.

Bergenstråhle, J., Bergenstråhle, L., & Lundeberg, J. (2020). SpatialCPie: an R/Bioconductor package for spatial transcriptomics cluster evaluation. *BMC bioinformatics*, 21(1), 1-7.

Berglund, E., Maaskola, J., Schultz, N., Friedrich, S., Marklund, M., Bergenstråhle, J., ... & Lundeberg, J. (2018). Spatial maps of prostate cancer transcriptomes reveal an unexplored landscape of heterogeneity. *Nature communications*, 9(1), 1-13.

Elosua-Bayes, M., Nieto, P., Mereu, E., Gut, I., & Heyn, H. (2021). SPOTlight: seeded NMF

regression to deconvolute spatial transcriptomics spots with single-cell transcriptomes. *Nucleic acids research*, 49(9), e50-e50.

Ståhl, P. L., Salmén, F., Vickovic, S., Lundmark, A., Navarro, J. F., Magnusson, J., ... & Frisén, J. (2016). Visualization and analysis of gene expression in tissue sections by spatial transcriptomics. *Science*, 353(6294), 78-82.

Reviewer #2:

Remarks to the Author:

This manuscript proposed a tool named MIST for annotating spots and imputing spatial expression values. I have some major concerns as below.

1) The first part of MIST for detecting functional region is based on the similarity graph after filtering low-weight edges. The authors only show its results compared with BayesSpace on one ST data from mouse brain sample, which is not sufficient. Moreover, BayesSpace is designed for subspot resolution, the authors didn't show details on their subspot annotation by MIST.

2) There are several methods developed specifically for subspot annotation, e.g. RCTD, SpatialDWLS, DSTG, SPOTlight, etc. The MIST's capability of annotation is not convincible.

3) The statement of "detecting functional region" is not clear to audience.

4) This manuscript spends most part on imputation rather than annotating, which may lose balance. More results are needed from the first part of MIST.

5) This imputation part adapts the low-rank-matrix completion algorithm via SVD similar with mcImpute. However, from the method section, this adaptation doesn't incorporate or consider the location information in spatial data, thus, it is difficult to convince me that it can be better or more suitable for imputation of spatial transcriptomics data.

6) Moreover, according to a systematic evaluation of single-cell imputation methods (PMID: 32854757), MAGIC and SAVER were shown outperform the mcImpute (etc) in most cases, which is different in this study. It may due to the limited (four) ST data and the small number of spots used in this manuscript. More and larger size of ST data are needed to compare the imputation performance.

7) Again, based on this study (PMID: 32854757), most imputation methods do not improve performance in downstream analyses compared to no imputation. Regarding the intra-cortex heterogeneity (Fig 4) and co-expression pattern (Fig 5), the authors may check if there are differences without using MIST.

8) There are methods designed specifically for spatial transcriptomics imputation, e.g. stPlus (PMID: 34252941), FIST (PMID: 33826608), and Tangram (PMID: 34711971), etc. The authors should benchmark with these methods instead of single-cell imputation methods.

Reviewer #3:

Remarks to the Author:

This research work explores a two-stage computational framework for region segmentation and gene expression imputation for spatial transcriptomics data. In the first stage, a co-expression graph is constructed with edges connecting neighboring spots and then the highly connected components are reported as clusters. In the second stage, gene expressions are imputed with sparse matrix completion regularized by nuclear norm. Below are the comments concerning the methodology and the evaluations.

1. While segmentation might help the following imputation stage in the cases where the

assumption of large connected regions holds, it is not clear why this stage is necessarily helpful in general. For example, the segmentation might result in very small fragmented regions, which are not informative for imputation. There could also be mixed tissues with sophisticated spatial arrangement. In Figure 4, some of purple spots (TH) are disconnected in some very small regions. It is unclear how the imputation can be meaningful for such small regions. It is important to justify/clarify the assumptions and applicability of this method.

2. It appears that the method is entirely based on McImpute except for python implementation and the focus on the segmented region? It should be articulated where the main difference is. In fact, a section "Related Methods" should be added to describe the baseline methods and how they are applied to the datasets.

3. There are many other methods for segmenting ST data (and/or H&E staining image) into regions based on clustering, which does not require continuity of the spots in the same cluster, even if the clusters are often naturally connected regions due to the high co-relation in nearby spots. What if these clustering methods are used in the first stage?

4. The motivation of region augmentation is unstated. Other than numerical consideration, this procedure does not seem to be straight-forward for regions of different sizes and tissues of different level of heterogeneity. This is unexplored in the current work.

5. spKNN might not be the state of the art for spatial imputation applied to spatial transcriptomics data. Other better baselines should be compared.

6. The four datasets used in the experiment are not very well described. Why were the four datasets chosen for evaluating the method? Possibly, it is better to use datasets of more variety of spatial patterns for the evaluation.

7. Some of the wordings are not exactly accurate, for example the subtitle "Graph embedding" is misleading since the paragraph is about graph construction. There is no "embedding" at all.

Reviewer #4:

Remarks to the Author:

Summary

This interesting paper by Wang and Liu developed a tool, MIST, for the analysis of spatial transcriptomics datasets. MIST has two objectives: (1) automatically define regions and boundary within a section; (2) impute for missing data in the gene expression matrix. The authors also showed some impressive applications of MIST, such as it helps resolve the structure within the data, and improved gene-gene co-expression signals.

Overall, this is a very useful tool to analyze 10X spatial transcriptomics data. The notebooks provided on the GitHub repository is also helpful for users to reproduce the analysis. However, in some part of the manuscript, the methods are not well explained and need clarification.

Major comments

- It is unclear to me how does MIST automatically find the threshold ϵ used for edge removal. Under method section "Edge removal and parameter selection", the authors said defined how the RMSE is calculated, and said, "X" represents the MIST denoised gene expression matrix using certain ϵ and n denotes the number of non-zero hold-out values. However, ϵ is not described in RMSE. Is the selection of ϵ also depend on the number of isolated spots?
- It seems to me that the imputation algorithm is developed based on McImpute. In the result section where the algorithm was described, the authors should clearly describe what improvements MIST has over McImpute on imputation.
- Under the method section "Low-rank matrix completion", the authors said, "Compared with McImpute, MIST also forces the observed values to be unchanged". The authors should provide rationale why this is preferred. Because if the assumption of drop out, is that transcripts are randomly not being captured, then the missed read can also happen to transcripts that is non-zero. If that is the case, then one would want to also "recover" the observed values as well.

Minor comments

- In Figure 2 where the authors used other imputation algorithms and compared their performance with MIST, MAGIC seemed to perform comparably to MIST and McImpute except for in the prostate dataset. Does the author have any intuition why MAGIC failed on the prostate dataset?
- In Figure 5, the authors demonstrated two examples where MIST recovered co-expressed pair of genes. In the text, the authors should explain why this is biologically meaningful. Additionally, negative examples should also be included to demonstrate the imputation step will not incorrectly introduce unmeaningful co-expressions.
- In the discussion, the authors should also include limitations of MIST. For example, automated boundary detection does not appear perfect, as many spots do not belong to a specific boundary (Figure 2. c&e).

All four reviewers provided very thoughtful reviews and constructive suggestions, which we appreciate and have accommodated to the best of our ability. Below, for the sake of convenience, we italicize the original reviewer comments and place our responses in regular roman font.

Reviewer #1

Spatial transcriptomics developed in 2016 by Ståhl et al. (2016) and allows in situ capturing of transcriptome-wide gene expression on almost single-cell resolution, is now a mature technology. Similar to single-cell sequencing technologies some challenges remain in performing improved downstream analyses of ST data; these are mainly variation in gene expression caused by the efficiencies of the cDNA library formation and sequencing, technical dropouts, and batch effects. The application of normalisation techniques tries to address these problems. Another important part of ST data analysis is the identification of regions having a unique gene expression or gene expression profile.

The authors of the manuscript propose a method, called Missing-value imputation and in silico region detection for spatially resolved transcriptomics (MIST), to improve ST data quality by removing technical dropouts within regions of similar gene expression. The method first identifies regions with similar gene expression using a 2D graph after edge removal if such a region contains more than five spots, and second, it removes technical dropouts within such a region using iteratively low-rank matrix completions. The authors claim that MIST accurately recovers spatial transcriptomics missing values, identifies intra-tissue heterogeneity and recovers spatial gene-gene co-expression signals.

Major critic points

1)Manual Annotation

The authors criticise that tissue sections that were analysed with ST still require manual annotation by scientists, e. g. pathologists, to obtain knowledge about the cell type-specific structure within a tissue sample. However, even if applying the MIST method or any other clustering or dimension reduction method, the resulting regions have to be annotated by a scientist as these methods does not offer information, which cell type or gene expression profile was captured; only the spatial distribution of regions with similar gene expression is shown. To provide a method that indeed helps to annotate regions having similar gene expression or even obviates the work of an annotator, more information for each identified region needs to be provided.

Thank you for pointing out the limitation on the previous version of MIST. We agree that it would be great if biologically meaningful region names could be automatically assigned to each detected region. While MIST cannot fully automate this process, it greatly facilitates the annotation by providing regional markers and enriched pathways/gene ontology terms associated with each region (**Fig. 2d, e**). We have now added such functionality to the MIST software so that users can run the analysis directly, view the regional markers, visualize the enrichment analysis results, and integrate their prior knowledge to name the regions. We have also provided a tutorial on how to perform such analysis on our GitHub repository.

2)Spatially connected regions of similar gene expression

The authors state that there is no method that addresses “the region-detection task in a way that establishes the boundaries of different tissue regions”. Spatial Transcriptome Decomposition (STD) developed for ST data and used in Berglund et al (2018) to identify cancer, PIN3, inflammation, healthy epithelial, and stroma regions in multiple tissue sections and Spotlight (Elosua-Bayes et al., 2021) are at least two methods that were developed for ST data and address the identification of regions with a unique gene expression profile. Both also normalises ST data and removes technical dropouts. Another example of a clustering method especially developed for ST data is SpatialCPie (Bergensträhle et al, 2020), which

is applied to human heart tissue samples. It is recommended that the authors compare MIST with these methods.

The reviewer suggests we compare MIST with STD¹, Spotlight², and SpatialCPie³. Unfortunately, these methods are not really comparable to MIST because they serve different functions. STD and Spotlight are a class of deconvolution methods that decompose the cell types or factors in the underlying tissue, while MIST's region detection is aimed at spatial clustering. SpatialCPie's main function is to visualize the correlations among clusters at different resolutions, e.g., the number of clusters used in K-Means.

To benchmark MIST and more fairly comparable methods, we have now compared MIST with other well-known clustering methods, namely, Louvain^{4,5}, Leiden⁴, K-Means⁶, and Hierarchical Clustering⁷.

3) Mixed cell types within tissue sections

The proposed method identifies spots having similar gene expression. However, if a tissue section contains mixed cell types besides regions of unique cell types, for example, a region of epithelial cells, a region of stromal cells, and a region of a mix of epithelial and stromal cells within one tissue sample, how does the proposed method address this?

MIST detects regions with highly similar cell types or mix of cell types. Therefore, regions with a mixture of cell types will be grouped into a region if the mix proportions are similar across these regions. For example, the stromal region (red) in **Figure 2** is a region with multiple cell types.

If, on the other hand, the proportions in the mix are highly variable across these spots, our algorithm will leave them as isolated spots. For example, the gray spots between the black (tumor) region and red (stroma) region in **Figure 2b** could be treated as a tumor microenvironment where tumor cells meet with stromal cells.

4) Tissue examples

The authors apply the MIST method to three examples for which Spatial Transcriptomics was applied to. However, the mouse brain samples dominate the manuscript. To provide insights into the generalizability of the MIST method, it has to be applied to a broader number of ST tissue samples. For example, the gene expression distribution of only one gene is shown in only one tissue sample of the prostate cancer example comprising twelve tissue samples and up to ~9000 transcripts per spot.

We have now added nine publicly available datasets from Visium's website (**Supp. Table 1**) to the original four datasets. These thirteen datasets varied in the number of spots, genes and sparsity levels, making the evaluation more inclusive and conclusion more generalizable. We benchmarked both MIST's region detection and imputation performance with other approaches using all of these data sets. We demonstrated MIST's efficacy for each functionality in the revised **Figures 2 and 3** respectively.

5) Multiple tissue samples

Although the authors do not claim that the MIST method can be applied to multiple tissue sections it can be a major part in analysing tissue sections. Then, batch effects can influence the analysis of multiple tissue samples. There is a risk when using MIST for multiple samples, that similar regions across tissue sections are missed. Can MIST applied to process multiple tissue sections? If not, this might be a major limitation of the proposed method.

MIST is designed to run on single ST data for region detection and imputation. Joint calling from multiple samples was not in our design. However, batch correction can be done using independent algorithm either before or after running MIST, since MIST does not borrow information from other samples.

6) Comparison to other methods

A major critical point is that the variance of gene expression due to technical problems in library formation and sequencing is neglected as well as batch effects if multiple tissue samples are analysed. Both problems are usually addressed using normalisation methods available for ST data and single-cell sequenced RNA (the latter can also be applied and is often used to analyse ST data). Therefore, the most common order in pre-processing ST data and preparing for downstream analysis is firstly normalisation including removal of batch effects, technical dropouts and variance in gene expression due to technical challenges (across multiple tissue samples) and secondly identification of regions (across similar tissue section) having similar gene expression or similar gene expression profiles.

Although normalisation across multiple tissue sample might remove biological meaningful differences. To be able to compare the performance of the proposed method, the result should be compared with results for which both aforementioned steps (normalisation, clustering/dimension reduction) were applied, for example, using the regions identified in Berglund et al (2018) using STD. Further, it is recommended to use normalisation methods for a comparison that are very common for scRNA, for example BASiCS, SCnorm, and scran (Lytal et al., 202) although they consider not only technical dropouts.

These are good points, and we have added comparisons between MIST and other traditional clustering methods used in single-cell RNA-seq analysis (K-Means, Leiden, Louvain, and Hierarchical clustering). As noted above, however, STD is not really comparable to MIST because it aims to decompose the cell types or factors in the underlying tissue while MIST aims at spatial clustering. In Berglund et al (2018)¹, the regions are human expert-defined pathological tissue domains, which are not spatial clusters automatically detected from one ST sample.

We agree that normalization is important for single-cell and spatial transcriptome analysis. Before running MIST, we followed the normalization procedure as instructed by ScanPy⁸. We have added the preprocessing details in the method section.

7) Just by eye, the overlap of the pathologist's annotation and the MIST regions shown in Figure 2d and 2f is not very convincing although higher than for the BayesSpace regions shown in Fig 2f.

Thank you for this comment— we have now quantified the clustering performance using the Silhouette coefficient, which gives us confidence that MIST is indeed better than other methods (now **Fig. 2f**). (Note that Fig. 2d is now **Fig. 2b** and 2f is now **Supp. Fig. 2c**.)

8) Benchmarking

The authors use the adjusted Rand index (ARI) to benchmark the MIST method. Although ARI is an appropriate and common benchmark for clustering, it would be helpful to use additional benchmarks to compare the clustered ST data for which alternative methods were applied, e.g. the Silhouette Coefficient.

Silhouette Coefficient was an excellent suggestion for an evaluation metric. As spot-level ground truth is not available to us for the majority of the samples we are testing, the Silhouette Coefficient is indeed the best available measurement metric available to us. Therefore, we added it to compare MIST's performance with other methods and added the results in **Figure 2f**.

Minor critic points

1) What is the statistical motivation to use a log fold change cut off?

Trusting a single p-value when assessing group differences can be biased by group size (p-values go down when the sizes of the groups increase⁹). To avoid such bias, we can apply an additional filtering layer on the effect size (fold-change or log-fold-change). Such approaches are used by well-known packages aiming at detecting differentially expressed genes, such as DESeq2¹⁰ and Limma¹¹. Using log-

fold-change allows us to understand the direction of the differential expression more intuitively since a negative value represents down-regulation while a positive value means up-regulation. In this manuscript, a cutoff of 0.26 on log-fold-change indicates a fold-change of 1.5 while a cutoff of 0.59 allows us to detect genes with no less than two-fold change.

References cited in our response above:

- Lytal, N., Ran, D., & An, L. (2020). Normalization methods on single-cell RNA-seq data: an empirical survey. *Frontiers in genetics*, 11, 41.
- Bergensträhle, J., Bergensträhle, L., & Lundeberg, J. (2020). SpatialCPie: an R/Bioconductor package for spatial transcriptomics cluster evaluation. *BMC bioinformatics*, 21(1), 1-7.
- Berglund, E., Maaskola, J., Schultz, N., Friedrich, S., Marklund, M., Bergensträhle, J., ... & Lundeberg, J. (2018). Spatial maps of prostate cancer transcriptomes reveal an unexplored landscape of heterogeneity. *Nature communications*, 9(1), 1-13.
- Elosua-Bayes, M., Nieto, P., Mereu, E., Gut, I., & Heyn, H. (2021). SPOTlight: seeded NMF regression to deconvolute spatial transcriptomics spots with single-cell transcriptomes. *Nucleic acids research*, 49(9), e50-e50.
- Ståhl, P. L., Salmén, F., Vickovic, S., Lundmark, A., Navarro, J. F., Magnusson, J., ... & Frisén, J. (2016). Visualization and analysis of gene expression in tissue sections by spatial transcriptomics. *Science*, 353(6294), 78-82.

Reviewer #2

This manuscript proposed a tool named MIST for annotating spots and imputing spatial expression values. I have some major concerns as below.

1) The first part of MIST for detecting functional region is based on the similarity graph after filtering low-weight edges. The authors only show its results compared with BayesSpace on one ST data from mouse brain sample, which is not sufficient. Moreover, BayesSpace is designed for subspot resolution, the authors didn't show details on their subspot annotation by MIST.

MIST is not designed to divide the spot into sub-spot resolution, so we could not compare MIST with BayesSpace at such a resolution. As BayesSpace detects the spatial clusters in the first step before making the sub-spot resolution, we compared with their spot-level performance.

We have, however, added comparisons between MIST and other well-known clustering methods: K-Means, Leiden, Louvain, and Hierarchical Clustering (**Figure 2f**).

2) There are several methods developed specifically for subspot annotation, e.g. RCTD, SpatialDWLS, DSTG, SPOTlight, etc. The MIST's capability of annotation is not convincing.

RCTD, SpatialDWLS, DSTG, and SPOTlight are cell-type deconvolution methods designed to decompose the cell type proportions at every spot. Unlike these methods, MIST's region detection focuses on automatically circling out major regions in the tissue, facilitating downstream analysis such as detecting regional DEGs (**Fig. 2d**) and gene set enrichment analysis (**Fig. 2e**). Therefore, MIST's region detection functionality is different from these mentioned methods, and we don't think they are comparable.

3) The statement of "detecting functional region" is not clear to audience.

We apologize for causing confusion. We were trying to express the concept that each region MIST detected contains functionally similar spots. We have removed the term 'functional' in the manuscript.

4) *This manuscript spends most part on imputation rather than annotating, which may lose balance. More results are needed from the first part of MIST.*

We added more evaluations on the first part of MIST's region detection (Method, section 'Benchmarking region detection accuracy'). Specifically, we evaluated MIST's spatial clustering (region detection) performance in Silhouette coefficients using both transcriptomes and spatial information (**Fig. 2f**). Moreover, we benchmarked MIST with more well-known clustering methods with an additional nine public datasets from 10X Visium (**Supp. Table 1**).

5) *This imputation part adapts the low-rank-matrix completion algorithm via SVD similar with mcImpute. However, from the method section, this adaptation doesn't incorporate or consider the location information in spatial data, thus, it is difficult to convince me that it can be better or more suitable for imputation of spatial transcriptomics data.*

MIST makes use of spatial information in an indirect way. MIST's imputation is a region-based mini-batch matrix-completion algorithm, where each region goes through multiple runs of matrix completion with random spots from outside the region (**Supp. Algorithm 1-3**). The regions detected at the first step does take spatial information into account, because it constructs the graph by connecting only spatially adjacent spots.

As shown in **Figure 3**, MIST outperformed McImpute by achieving a higher Pearson Correlation Coefficient and lower Rooted Mean Square Error when recovering hold-out gene expression values. This result confirms that using region-based imputation improved the imputation performance.

6) *Moreover, according to a systematic evaluation of single-cell imputation methods (PMID: 32854757), MAGIC and SAVER were shown outperform the mcImpute (etc) in most cases, which is different in this study. It may due to the limited (four) ST data and the small number of spots used in this manuscript. More and larger size of ST data are needed to compare the imputation performance.*

As noted above, we collected nine additional Visium datasets from 10X's website (**Supp. Table 1**). The results showed that MIST outperformed all other methods compared. In Hou et. al. (PMID: 32854757)¹², we observed that all three methods were ranked high, which is the major reason we benchmarked MIST against them.

Regarding the discordant performance order of McImpute, MAGIC, and SAVER, there could be several causes. First, the datasets differ due to platform differences, e.g., single-cell datasets have single-cell resolution, and there are different sequencing depths. Second, we used a different evaluation approach than Hou et al¹², which benchmarked performance by comparing the downstream analysis, such as comparing the similarity with the bulk data. However, we used hold-out experiments to simulate random hold-out and assess imputation methods' ability to recover the hold-out values.

7) *Again, based on this study (PMID: 32854757), most imputation methods do not improve performance in downstream analyses compared to no imputation. Regarding the intra-cortex heterogeneity (Fig 4) and co-expression pattern (Fig 5), the authors may check if there are differences without using MIST.*

To check if there are differences without using MIST, we first extracted the marker genes between the two spatial clusters detected by MIST and then examined these genes' behaviors in the original data, without imputation. Since the raw data is sparse and will introduce errors in detecting the true signals, for each gene we examined only cluster-specific non-zero gene expression values. For quality control purposes, we examined only genes that are expressed in at least five spots in both clusters. Then, we calculated the LFC on these non-zero gene expression values. By doing so, we provide a pseudo-ground-truth to compare with (as there is no actual ground-truth).

We first show that the LFCs calculated on the non-zero spots are not consistent with the LFCs calculated on all cluster-specific spots with the original data (left figure), where we not only see the different enrichment degrees but also changes of directions. Then, when comparing the LFCs calculated on the non-zero spots with LFCs calculated on all cluster-specific spots with MIST imputed data (right figure), this deviance was corrected, and the points were aligned at the $y = x$ diagonal.

We believe that such heterogeneity was hidden in the original data even without MIST imputation. Because of imputation, MIST is able to expose such hidden differences.

To further validate the co-expression patterns of the two pairs of genes, namely *Cldn11*-*Arhgef10* and *Gfap*-*Aqp4* in mouse brain, we examined another single-cell RNA-sequencing dataset published by Methodios Ximerakis in 2019 (GSE129788)¹³. 16 mouse brains were sc-RNA sequenced in the study. We examined the co-expression at both the single-cell level and a pseudo-bulk level, which takes the average gene expression for each mouse brain to reduce the effect of dropout. As shown in the figure below (also **Supp. Fig. 5.1**), these two pairs of genes show significant positive correlations (P-value = 0) at both single-cell level and pseudo-bulk level, demonstrating the biological validity of the co-expression patterns recovered by MIST.

8) There are methods designed specifically for spatial transcriptomics imputation, e.g. stPlus (PMID: 34252941), FIST (PMID: 33826608), and Tangram (PMID: 34711971), etc. The authors should benchmark with these methods instead of single-cell imputation methods.

We appreciate the rationale for suggesting we compare with stPlus, FIST, and Tangram, but unfortunately, they are not comparable imputation methods for the following reasons.

1. stPlus estimates expression values for genes that are not measured in spatial transcriptomics, while MIST estimates expression values for genes that are measured. Moreover, stPlus requires scRNA-seq profile data from the same tissue, and the method was tested only on FISH-based datasets. These platforms depend on a small number of pre-designed probes. In contrast, our method was developed to address the dropout issues in the 10x Visium platform and reduce the noise on measured genes. These two methods are therefore doing entirely different things.
2. Tangram is an sc/snRNA sequencing to spatial transcriptomics mapping algorithm that maps single cells (nucleus) to the spatial locations from the same tissue. The corrected (imputed) gene expression values are predicted by the mapped single cells (nucleus) rather than the spatial transcriptomics. Tangram requires sc/sn-RNA sequencing of the same tissue and sample to perform the spatial gene expression prediction function, which is costly and therefore rarely available. For example, all the data sets **Supplementary Table 1** have no accompanying sc/sn-RNA sequencing data available. Thus, we are not able to compare Tangram.
3. FIST is the most similar approach to MIST's imputation function to estimate the dropout values in microarray-based spatial transcriptomics, such as Visium. Unfortunately, FIST's code is not well-organized, and the scripts are put in a repository without instructions on how they can be used on new datasets (<https://github.com/kuanglab/FIST>). They provided scripts to reproduce the results in their manuscript¹⁴, e.g., cross-validation experiments, but do not allow user-friendly application of their method on new datasets. We have been unable to figure out how to apply this method.

We have therefore been unable to find a method that uses spatial imputation to benchmark with MIST.

Reviewer #3

This research work explores a two-stage computational framework for region segmentation and gene expression imputation for spatial transcriptomics data. In the first stage, a co-expression graph is constructed with edges connecting neighboring spots and then the highly connected components are reported as clusters. In the second stage, gene expressions are imputed with sparse matrix completion regularized by nuclear norm. Below are the comments concerning the methodology and the evaluations.

1. While segmentation might help the following imputation stage in the cases where the assumption of large connected regions holds, it is not clear why this stage is necessarily helpful in general. For example, the segmentation might result in very small fragmented regions, which are not informative for imputation. There could also be mixed tissues with sophisticated spatial arrangement. In Figure 4, some of purple spots (TH) are disconnected in some very small regions. It is unclear how the imputation can be meaningful for such small regions. It is important to justify/clarify the assumptions and applicability of this method.

In brief, the isolated spots will be jointly imputed with spots in detected regions. Indeed, these points

will be challenging for imputation as compared to spots in well-connected regions. To address this issue, we imputed these isolated spots multiple times using different regions and subsampled auxiliary spots. Then, we used ensemble approach to combine estimates from different sub-samplings. The detail is in **Supplementary Algorithm 3**.

2. It appears that the method is entirely based on McImpute except for python implementation and the focus on the segmented region? It should be articulated where the main difference is. In fact, a section "Related Methods" should be added to describe the baseline methods and how they are applied to the datasets.

The relationship between MIST and McImpute is like the relationship between Random Forest and Decision Tree, with MIST being the ‘Random Forest’ and McImpute the ‘trees’. McImpute achieves a low-rank approximation given a set of samples and genes. We used mini batch-based ensemble methods that take the low-rank completion as the baseline, and then aggregated individual batches to make the final predictions (**Supp. Algorithm 3**).

Ensemble methods such as Random Forest and other mini-batch-based methods in the bioinformatics area, such as *mbkmeans*¹⁵, have proven to be both accurate (by reducing overfitting) and computationally efficient. We demonstrated that MIST outperforms McImpute in the hold-out experiments (**Figure 3**).

3. There are many other methods for segmenting ST data (and/or H&E staining image) into regions based on clustering, which does not require continuity of the spots in the same cluster, even if the clusters are often naturally connected regions due to the high co-relation in nearby spots. What if these clustering methods are used in the first stage?

This is a great point. The idea is to use a clustering approach to reduce dimensionality and provide biological priors for mini-batch-based matrix completion. Any good clustering algorithm could replace our segmentation and produce similar or even better results. However, in this paper, since we demonstrated that MIST’s region detection preserves better spatial and transcriptome-wise clustering structures in **Figure 2f**, our imputation part will proceed with the spatial clustering (region detection) results.

4. The motivation of region augmentation is unstated. Other than numerical consideration, this procedure does, not seem to be straight-forward for regions of difference sizes and tissues of different level of heterogeneity. This is unexplored in the current work.

The motivation is borrowed from ensemble approaches like Random Forest. Whereas McImpute achieves a low-rank approximation given a set of samples and genes, MIST uses mini batch-based ensemble methods that take the low-rank completion as the baseline, and then individual batches are aggregated to make the final predictions (**Supp. Algorithm 3**). We added the motivation and intuition in the Method section, ‘*Imputation using region-based mini-batch matrix completion*’ subsection.

5. spKNN might not be the state of the art for spatial imputation applied to spatial transcriptomics data. Other better baselines should be compared.

To our knowledge, spKNN is the only one used by other ST studies. For example, He, et., al, 2020¹⁶, used spKNN to smooth the top variable genes and build a deep learning method to predict spatial gene expression values on these spKNN-imputed values.

Another relevant method is FIST¹⁴, which applies a network-guided tensor completion to impute spatial transcriptomics. However, we were not able to run their pipeline using our dataset due to insufficient instructions and unorganized scripts on their GitHub repository (<https://github.com/kuanglab/FIST>). However, they also used spKNN as a baseline spatial-aware

imputation method to compare against in their manuscript¹⁴.

6. *The four datasets used in the experiment are not very well described. Why were the four datasets chosen for evaluating the method? Possibly, it is better to use datasets of more variety of spatial patterns for the evaluation.*

To make our comparison more generalizable, we added nine additional samples to compare both region detection and imputation (**Supp. Table 1**). These datasets include mostly human samples from different tissues and vary in number of spots, genes, and the sparsity levels.

7. *Some of the wordings are not exactly accurate, for example the subtitle "Graph embedding" is misleading since the paragraph is about graph construction. There is no "embedding" at all.*

This is a challenge of being non-native speakers! Thank you for pointing this out; we have changed ‘Graph embedding’ to ‘Graph construction’.

Reviewer #4

Summary

This interesting paper by Wang and Liu developed a tool, MIST, for the analysis of spatial transcriptomics datasets. MIST has two objectives: (1) automatically define regions and boundary within a section; (2) impute for missing data in the gene expression matrix. The authors also showed some impressive applications of MIST, such as it helps resolve the structure within the data, and improved gene-gene co-expression signals.

Overall, this is a very useful tool to analyze 10X spatial transcriptomics data. The notebooks provided on the GitHub repository is also helpful for users to reproduce the analysis. However, in some part of the manuscript, the methods are not well explained and need clarification.

Major comments

• *It is unclear to me how does MIST automatically find the threshold ϵ used for edge removal. Under method section “Edge removal and parameter selection”, the authors said defined how the RMSE is calculated, and said, “ X ’ represents the MIST denoised gene expression matrix using certain ϵ and n denotes the number of non-zero hold-out values. However, ϵ is not described in RMSE. Is the selection of ϵ also depend on the number of isolated spots?*

We apologize for not being clear. To put it simply and make the parameter selection automatic and efficient, we have changed our parameter selection step and updated it accordingly in the manuscript (**Method, Supp. Algorithm 2**). In brief, we used a grid search approach by maximizing intra-region similarities while minimizing inter-region similarities and the fraction of isolated spots (Equation 1 in the manuscript).

• *It seems to me that the imputation algorithm is developed based on McImpute. In the result section where the algorithm was described, the authors should clearly describe what improvements MIST has over McImpute on imputation.*

Thank you for this suggestion, we have added to our result section (**line 82-87**). While McImpute achieves a low-rank approximation using all the spots in the tissue sample, MIST uses mini batch-based ensemble methods that take low-rank completion as the baseline, and then aggregates individual batches to make the final predictions (**Supp. Algorithm 3**).

Ensemble methods such as Random Forest have proven to be both computationally efficient and more

accurate than single classifier (Decision Tree) because they reduce overfitting by the votes from multiple experts. Such minibatch-based approaches are especially beneficial for operating on high dimensional data, as it significantly reduces the time and memory requirement, which usually are exponentially increased regarding the number of features and sample size.

- *Under the method section “Low-rank matrix completion”, the authors said, “Compared with McImpute, MIST also forces the observed values to be unchanged”. The authors should provide rationale why this is preferred. Because if the assumption of drop out, is that transcripts are randomly not being captured, then the missed read can also happen to transcripts that is non-zero. If that is the case, then one would want to also “recover” the observed values as well.*

Excellent point, and this is now an option in our algorithm so one can choose to impute on both zero and non-zero values. Our examples are focused on zero elements since these are the most difficult cases, and hold-out experiments in imputation also use zeros as default.

Minor comments

- *In Figure 2 where the authors used other imputation algorithms and compared their performance with MIST, MAGIC seemed to perform comparably to MIST and McImpute except for in the prostate dataset. Does the author have any intuition why MAGIC failed on the prostate dataset?*

The differences in the performances among all the models are smaller in the Prostate data compared with other data sets. The minimum Pearson Correlation Coefficient (PCC) is 0.66 and maximum is 0.72 with a range of 0.06 for Prostate data. By contrast, in the Melanoma sample, the minimum PCC is 0.57 and maximum PCC is 0.81 with a range of 0.24.

We would therefore argue that MAGIC did not fail on the Prostate dataset, because it achieved a median Pearson Correlation Score of 0.67 and RMSE of 0.86. When comparing across samples, these numbers are competitive.

- *In Figure 5, the authors demonstrated two examples where MIST recovered co-expressed pair of genes. In the text, the authors should explain why this is biologically meaningful. Additionally, negative examples should also be included to demonstrate the imputation step will not incorrectly introduce unmeaningful co-expressions.*

In the first paragraph of section “MIST recovers spatial gene-gene co-expression patterns”, we explained the biological importance of recovering co-expression relationships between genes as follows: “Dropouts within ST datasets weaken the correlation analysis and cause inaccurate estimation of gene-gene correlation, which is the fundamental element in many analyses such as weighted correlation network analysis (WGCNA)”.

Regarding negative examples, we showed two additional gene pairs, *Gfap-Calm2* and *Gfap-Gng2*.

Gfap-Calm2 showed a negative spatial correlation of -0.55 in Allen Brain Atlas (**Supp. Fig. 5.2a**) and a significant negative correlation of -0.7 using the external single-cell mouse brain cohort (**Supp. Fig. 5.2h**). However, in the raw mouse brain ST data, we observed only a small negative correlation of -0.11 (**Supp. Fig. 5.2 b**). After imputation, MIST corrected the pattern by achieving a negative correlation of -0.37.

Gfap-Gng2 showed an insignificant spatial correlation of 0.19 in Allen Brain Atlas (**Supp. Fig. 5.3a**, p-value=0.57) and an insignificant correlation of 0.29 (p-value=0.28) using the external single-cell mouse brain cohort (**Supp. Fig. 5.3h**). Both the raw and imputed mouse brain ST samples show very low correlation scores with raw sample of 0.03 and imputed sample of 0.1. Therefore, MIST didn’t introduce unmeaningful co-expressions for such gene pairs.

- *In the discussion, the authors should also include limitations of MIST. For example, automated*

boundary detection does not appear perfect, as many spots do not belong to a specific boundary (Figure 2. c&e).

Thank you for the suggestion. We have discussed the isolated spots and their potentials in the discussion section, **line 247-250**.

References

1. Berglund, E. *et al.* Spatial maps of prostate cancer transcriptomes reveal an unexplored landscape of heterogeneity. *Nat. Commun.* **9**, 1–13 (2018).
2. Elosua-Bayes, M., Nieto, P., Mereu, E., Gut, I. & Heyn, H. SPOTlight: seeded NMF regression to deconvolute spatial transcriptomics spots with single-cell transcriptomes. *Nucleic Acids Res.* **49**, e50–e50 (2021).
3. Bergensträhle, J., Bergensträhle, L. & Lundberg, J. SpatialCPie: an R/Bioconductor package for spatial transcriptomics cluster evaluation. *BMC Bioinformatics* **21**, 1–7 (2020).
4. Traag, V. A., Waltman, L. & Van Eck, N. J. From Louvain to Leiden: guaranteeing well-connected communities. *Sci. Rep.* **9**, 1–12 (2019).
5. Levine, J. H. *et al.* Data-driven phenotypic dissection of AML reveals progenitor-like cells that correlate with prognosis. *Cell* **162**, 184–197 (2015).
6. Likas, A., Vlassis, N. & Verbeek, J. J. The global k-means clustering algorithm. *Pattern Recognit.* **36**, 451–461 (2003).
7. Murtagh, F. & Legendre, P. Ward’s hierarchical clustering method: clustering criterion and agglomerative algorithm. *arXiv Prepr. arXiv1111.6285* (2011).
8. Wolf, F. A., Angerer, P. & Theis, F. J. SCANPY: large-scale single-cell gene expression data analysis. *Genome Biol.* **19:15**, 2926–2934 (2018).
9. Gómez-de-Mariscal, E. *et al.* Use of the p-values as a size-dependent function to address practical differences when analyzing large datasets. *Sci. Rep.* **11**, 1–13 (2021).
10. Love, M. I., Huber, W. & Anders, S. Moderated estimation of fold change and dispersion for RNA-seq data with DESeq2. *Genome Biol.* **15**, 1–21 (2014).
11. Ritchie, M. E. *et al.* limma powers differential expression analyses for RNA-sequencing and microarray studies. *Nucleic Acids Res.* **43**, e47–e47 (2015).
12. Hou, W., Ji, Z., Ji, H. & Hicks, S. C. A systematic evaluation of single-cell RNA-sequencing imputation methods. *Genome Biol.* **21**, 1–30 (2020).
13. Ximerakis, M. *et al.* Single-cell transcriptomic profiling of the aging mouse brain. *Nat. Neurosci.* **22**, 1696–1708 (2019).
14. Li, Z., Song, T., Yong, J. & Kuang, R. Imputation of spatially-resolved transcriptomes by graph-regularized tensor completion. *PLoS Comput. Biol.* **17**, e1008218 (2021).
15. Hicks, S. C., Liu, R., Ni, Y., Purdom, E. & Risso, D. mbkmeans: Fast clustering for single cell data using mini-batch k-means. *PLoS Comput. Biol.* **17**, e1008625 (2021).
16. He, B. *et al.* Integrating spatial gene expression and breast tumour morphology via deep learning. *Nat. Biomed. Eng.* 1–8 (2020).

Reviewers' Comments:

Reviewer #1:

Remarks to the Author:

I appreciate that the number of tissue samples was increased for the hold-out experiment. The manuscript and the MIST algorithm improved. However, some of my concerns were not addressed:

Annotation

Throughout the manuscript the term "annotation" is misinterpreted. The authors repeatedly write that MIST annotates regions (automatically). If the algorithm only facilitates the process of annotation, as stated in the reviewer reply, please correct this throughout the manuscript. For example, in the Abstract it is stated that spatially resolved transcriptomics "must correctly annotate tissue regions". This is incorrect in my opinion; please reformulate or provide a reference. Please also specify which gene expression profiles MIST produces or reformulate if MIST only facilitates this in down stream analyses like other normalization methods for ST data do.

Fig 2d, please state clearly in the Figure text, which regions are compared in the volcano plots. None of the pathways enriched in the cancer region are linked to cancer. Please discuss this, and also how this regions could be identified as a cancer region because the authors state that MIST facilitates annotation of the connected regions. The 5 mentioned up regulated genes provide only limited information, e.g. NDRG1 is a tumour suppressor gene but up regulated in the cancer region.

A problem of the improved MIST algorithm is that pathway enrichment (GSEA) is suggested to be used for the functional annotation of regions. One cannot use enriched pathways to functionally annotate.

Spatially connected regions of similar gene expression

The authors emphasise that MIST is able to identify meaningful cell clusters in tissue samples. The provided result of connected regions for the mouse brain is not sufficient to generalise. Please provide for additional tissue sections the regions of similar gene expression identified by the MIST algorithm (if possible incl DGE analysis), even if a pathological annotation on spot level for these additional samples is not available. If possible compare the MIST-regions to connected regions identified by other algorithms.

Intra-patient heterogeneity

Similar for the increased heterogeneity experiment. As I understand, an functional annotation on spot level would be needed for this task, however, one provided sample (mouse brain) is not sufficient for a generalisation. Please add additional tissue sections to show that MIST improves the clustering results, also compared to other algorithms as the authors show for the mouse brain sample in Fig 4.

Limitations

A major limitation of the MIST is it is only applicable to single samples. The quality of the ST data will influence the granularity of the detected regions, i.e. an inter-sample comparison of identified regions is difficult. If the latter is wrong, please provide examples. Further, only tissue samples with a larger number of spots can be analysed with MIST. These limitations need to be discussed because it might be important for a user.

Minor comments

A fold change cutoff is arbitrary from a statistical point of view even if it is popular to apply it.

Table S1 please cite correctly.

Additional comments

Threshold for minimum #spots in a region is set to 20. Why was this value chosen ? Is it set by the user or optimised by the MIST algorithm ?

Line 45, could you provide an example of a region that is biologically distinct, not detectable by eye but was identified using MIST ?

Line 53, “.. technical dropouts can make the gene expression profile sparse and produce excessive zero values in the gene expression data”. I can’t find this statement in the cited paper (reference #8). Please double check.

Some statements seem to be

- exaggerated, like:

--Line 24, “MIST excelled”. Based on 1 sample, it is vague to claim this.

--Line 246, “greatly”. The difference in Fig 3a seem not to be great, in HBCB1 from ~ 0.9 (McImpute) to ~ 0.8 (MIST); in some sample the difference is even less.

- overcritical, like

--Line 264, “ST data provided by 10X Visium might hinder the accuracy of ST data analysis”. I think ‘hinder’ is not the right term.

--Line 49, “... none of them is able to establish the boundaries of different tissue regions.” Is this true ? What is shown in Fig S2c, clusters by BayesSpace, instead ?

Line 238, “We view these isolated spots as important elements for domain-specific studies, such as tumor microenvironments”. To merge all spots that could not be assigned to a cluster into one functional group seems imprecise.

Line 265, “dramatically improves the signal-to-noise ratio.” Where are the values of signal-to-noise-ratio shown ?

Reviewer #2:

Remarks to the Author:

Though the authors provide responses to my comments, I don’t think they address my major concerns.

1) The authors only compared with simple clustering methods such as KMeans, Leiden, Louvain, and Hierarchical Clustering. These methods are not comparable with MIST, as they are not well used to cluster spatial data. I do recommend comparing with the spatial designed methods, such as STAGATE (PMID: 35365632) and spaGCN (PMID: 34711970).

2) Moreover, the authors show the clustering performance based on the Silhouette index, which is insufficient and inappropriate, especially when the spatial data has ground truth. Though the authors include the 10x visium data, I would recommend doing the comparisons based on the 10x Visium dataset containing spatial expressions of 12 human dorsolateral prefrontal cortex (DLPFC) section (PMID: 33558695). The comparison results are preferred to be shown as figure with ARI metrics.

3) The authors argued that the cell-type deconvolutions are not comparable with MIST, however, I would like to check if the imputed data by MIST does improve the deconvolution performance? It is one way to evaluate if the imputation improves downstream analysis.

4) I would not agree with the authors’ comments saying they are unable to find a method that uses spatial imputation to benchmark with MIST, since the authors show that MIST outperformed McImpute by achieving a higher Pearson Correlation Coefficient (PCC) and lower Rooted Mean Square Error (RMSE) when recovering hold-out gene expression values. Could you add DeepImpute (PMID: 31627739) and make a figure on PCC and RMSE with each method (MIST, DeepImpute, McImpute, MAGIC, SAVER)?

Overall, I would recommend the authors to verify the performance of MIST at both imputation and clustering aspects, which are the two major components of this method, through meaningful comparisons.

Reviewer #3:

Remarks to the Author:

I have no more comments.

Reviewer #4:

Remarks to the Author:

The authors have addressed all my concerns to my satisfaction.

All four reviewers provided very thoughtful reviews and constructive suggestions, which we appreciate and have accommodated to the best of our ability. Below, for the sake of convenience, we italicize the original reviewer comments and place our responses in regular roman font.

Reviewer #1:

I appreciate that the number of tissue samples was increased for the hold-out experiment. The manuscript and the MIST algorithm improved. However, some of my concerns were not addressed:

1) Annotation

Throughout the manuscript the term “annotation” is misinterpreted. The authors repeatedly write that MIST annotates regions (automatically). If the algorithm only facilitates the process of annotation, as stated in the reviewer reply, please correct this throughout the manuscript. For example, in the Abstract it is stated that spatially resolved transcriptomics “must correctly annotate tissue regions”. This is incorrect in my opinion; please reformulate or provide a reference. Please also specify which gene expression profiles MIST produces or reformulate if MIST only facilitates this in down stream analyses like other normalization methods for ST data do.

Thank you for the suggestions and sorry for confusion. We might have not made the point clear - ‘MIST allows automatic region detection and facilitates annotation by providing the list of activated genes and enriched functional terms for each region’. We have changed the wording accordingly throughout the manuscript. Below are where we changed with reference to the manuscript:

- Line 16: “This is easier said than done: the technique must correctly annotate tissue regions and accommodate excessive zero values of gene expression due to high dropout rates” was replaced by “Highly sparse gene expression values due to dropout events and a lack of tools to facilitate automatic region detection and annotation make analyzing such data sets challenging.”
- Line 24: “excelled” was changed to “facilitated”
- Line 27: “provides unbiased region annotations” was changed to “provides unbiased region detections to facilitate annotations with the associated functional analyses”
- Line 90: “To demonstrate MIST’s efficacy in region detection and annotation” was changed to “To demonstrate MIST’s efficacy in region detection and how it helps annotation”
- Line 239: “MIST enables annotation” was changed to “MIST facilitates user to annotate”

MIST will not change the observed expression values and will only impute zero-values that were potentially affected by random dropouts based on a region-specific low-rank approximation algorithm. Users could specify to impute the expression values with which normalization approach, e.g., count per million (CPM) or log normalized gene expression. By imputing the gene expression values, MIST reduces the number of random zeros and better resembles the real distribution according to the hold-out experiments and external references (Fig. 2 g-h, Fig. 5).

2) Fig 2d, please state clearly in the Figure text, which regions are compared in the volcano plots. None of the pathways enriched in the cancer region are linked to cancer. Please discuss this, and also how this regions could be identified as a cancer region because the authors state that MIST facilitates annotation of the connected regions. The 5 mentioned up regulated genes provide only limited information, e.g. NDRG1 is a tumour suppressor gene but up regulated in the cancer region.

Sorry for not making the figure caption clear, we have added the compared regions for each of the volcano plots (line 606).

We were using Gene Ontology terms, which is comprehensive but not complete. In our case, among the top up-regulated genes for the black (tumor) region, SFRP1 was reported to play important role in skin cancer initiation (Baena-Acevedo, Juvenal Darío, et al.); ATP1A1, also known as alpha-1 subunit of the sodium potassium ATPase, is reported to be an oncogene for Melanoma (Mathieu, Veronique et al.); SSP1, which produces Osteopontin, was reported to be a melanoma prognostic marker and silencing Osteopontin inhibited the proliferation and invasion of Melanoma cells(Rangel, Javier, et al.; Kiss, Timea, et al.); Although NDRG1 is tumor suppressor, it was reported to be highly expressed in tumors and might be used for diagnostic purpose (Ellen TP, et al.). Also, in the pathological data from the Human Protein Atlas, 7 out of 9 melanoma samples showed high antibody stains for NDRG1 protein (link). Therefore, we believe that these highly activated genes are useful markers to help identify tumor regions in the underlying tissues.

References used in the response:

Baena-Acevedo, Juvenal Darío, et al. "SFRP1, Possible Biomarker in the Progression or Regression of Cervical Pre-neoplastic Lesions Associated with Human Papilloma Virus." Infectio 25.4 (2021): 270-275.

Mathieu, Véronique, et al. "The sodium pump $\alpha 1$ sub-unit: a disease progression-related target for metastatic melanoma treatment." Journal of cellular and molecular medicine 13.9b (2009): 3960-3972.

Rangel, Javier, et al. "Osteopontin as a molecular prognostic marker for melanoma." Cancer: Interdisciplinary International Journal of the American Cancer Society 112.1 (2008): 144-150.

Kiss, Timea, et al. "Silencing osteopontin expression inhibits proliferation, invasion and induce altered protein expression in melanoma cells." Pathology and Oncology Research (2021): 2.

Ellen TP, Ke Q, Zhang P, Costa M. NDRG1, a growth and cancer related gene: regulation of gene expression and function in normal and disease states. Carcinogenesis. 2008 Jan;29(1):2-8. doi: 10.1093/carcin/bgm200. Epub 2007 Oct 4. PMID: 17916902.

3) A problem of the improved MIST algorithm is that pathway enrichment (GSEA) is suggested to be used for the functional annotation of regions. One cannot use enriched pathways to

functionally annotate.

Thank you for your comment. We think this comment is related to comment 1. Yes, we agree that only using GSEA will not be enough to automatically annotate regions. Domain knowledge on the marker genes to match with our region-specific differential gene expression (DEG) analysis.

We have changed the tone throughout the manuscript to specify that GSEA is coupled with MIST and provide insights for users to annotate the regions (specific lines referred above). MIST will provide the activated gene list and the GSEA results to users as tables and figures, and users could interactively name their regions based on these results.

4) Spatially connected regions of similar gene expression

The authors emphasise that MIST is able to identify meaningful cell clusters in tissue samples. The provided result of connected regions for the mouse brain is not sufficient to generalise. Please provide for additional tissue sections the regions of similar gene expression identified by the MIST algorithm (if possible incl DGE analysis), even if a pathological annotation on spot level for these additional samples is not available. If possible compare the MIST-regions to connected regions identified by other algorithms.

Thank you for your suggestions. Per reviewer's request, we quantitatively compared MIST and other region detection and clustering methods on 13 samples (**Supp. Table 1**) with different number of spots in Figure 2f. We also provide the detected regions for each sample in Supplementary Figure 2. The differential gene list and enriched pathway/ontology list for every detected region were included in the supplementary files.

5) Intra-patient heterogeneity

Similar for the increased heterogeneity experiment. As I understand, an functional annotation on spot level would be needed for this task, however, one provided sample (mouse brain) is not sufficient for a generalisation. Please add additional tissue sections to show that MIST improves the clustering results, also compared to other algorithms as the authors show for the mouse brain sample in Fig 4.

Sorry for causing the confusion. We didn't claim that MIST always detect intra-tissue heterogeneity even if the tissue is relatively homogeneous. In Figure 5, we found intra-cortex heterogeneity only in the 12-month mouse brain with Alzheimer's disease (AD), but not in the wild-type 12-month mouse brain. Here we show two additional tissue sections where MIST-imputation revealed different degrees of intra-tissue heterogeneity.

In the figure below (also **Supp. Fig. 4.5**), we showed that intra-cortex heterogeneity shown in **Fig. 4** also exists in the 18-month AD mouse brain even without imputation (**panel B**), validating that the heterogeneity recovered by MIST in Figure 5. Moreover, we could see the separation between spatial clusters were clearer after imputation, with ENTI (red) and COM (orange) spots in the same side after imputation (**panel D**). However, such heterogeneity was not clear after MAGIC imputation (**panel C**).

Figure 1| UMAP of 18-month mouse brain with Alzheimer's disease before and after imputation. (A) Manual annotation. (B) UMAP without imputation. (C) UMAP with MAGIC imputation. (D) UMAP with MIST imputation.

The figure below shows a human dorsolateral prefrontal cortex sample (also **Supp. Fig. 4.6**), where MIST imputation did not detect larger heterogeneity rather than making the region WM visually more apart from other layers in the UMAP (**panel B, D**). However, MIST preserved the original structure and did not overly impute that might distort the clusters like other imputation methods (panel C).

Figure 2 | UMAP of a Human DLPFC sample (15672) before and after imputation. (A) Manual annotation. (B) UMAP without imputation. (C) UMAP with MAGIC imputation. (D) UMAP with MIST imputation.

6) Limitations

A major limitation of the MIST is it is only applicable to single samples. The quality of the ST data will influence the granularity of the detected regions, i.e. an inter-sample comparison of identified regions is difficult. If the latter is wrong, please provide examples. Further, only tissue samples with a larger number of spots can be analysed with MIST. These limitations need to be discussed because it might be important for a user.

Thank you for your comments on MIST's limitations. We agree that the currently MIST only supports single-tissue region detection and imputation, which is a limitation that we would like to work on in the next steps. We discussed this limitation in the revised manuscript (line 277-281).

We showed MIST works well on a variety of ST samples with different number of spots, even with tissue samples with less than 300 spots (Melanoma, **Supp. Table 1.**) in terms of both region detection (**Fig. 2f**) and imputation accuracy (**Fig. 3**).

Minor comments

1) A fold change cutoff is arbitrary from a statistical point of view even if it is popular to apply it.

Thank you for your comment. In addition to the fold change cutoff, we applied filtering on the adjusted P-value based on a non-parametric statistical approach – Wilcoxon Rank Sum Test.

2) *Table S1 please cite correctly.*

Thank you for your suggestion. Because most of the data sets in Supplementary Table 1 were not published in articles but available on 10X's website, we decided to provide addresses from where we downloaded them in the 'Source or reference' column.

Additional comments

1) *Threshold for minimum #spots in a region is set to 20. Why was this value chosen? Is it set by the user or optimised by the MIST algorithm?*

The threshold is chosen based on the resolution of the data and is set by the user. The previous Spatial Transcriptomics data has a relatively low resolution and therefore we expect regions with small number of spots in such data sets. For the most recent Visium data, we recommend using a larger threshold like 40 because we have higher resolution and more spots for each sample. Moreover, if the users believe that smaller regions are of interest, a smaller threshold is preferred.

2) *Line 45, could you provide an example of a region that is biologically distinct, not detectable by eye but was identified using MIST?*

In the figure below we show the example of a 12-month Mouse brain sample (MouseAD) used in the study. **Panel (A)** shows the H&E staining images where the level-one regions (**Panel C**) were easily detectable by eyes. However, without expertise and prior knowledge of the topology of mouse brain, it is very hard to purely tell the differences of some level-2 regions (**Panel D**), for example, RSP and PTL on the lower quadrant of the cortex region. In this example, MIST identified the differences between some level-2 regions including RSP, PTL and some others. Although not all level-two subregions were identified, MIST did improve a finer resolution region detection upon the level-one annotations which are more easily detected by eyes.

Figure 3| Mouse brain example. (A) H&E staining image. (B) regions detected by MIST. (C) Level-one spot annotations. (D) Level-two spot annotations.

3) Line 53, “.. technical dropouts can make the gene expression profile sparse and produce excessive zero values in the gene expression data”. I can’t find this statement in the cited paper (reference #8). Please double check.

In the paragraph seven, here is what the authors wrote “Furthermore, we compared spatial transcriptomics with the near-100% sensitivity of single-molecule fluorescent in situ hybridization in adjacent tissue sections. The sensitivity of spatial transcriptomics was 6.9 ± 1.5 % of single molecule fluorescent in situ hybridization (fig. S6).”. Since this is due to technical differences between sequencing-based Spatial Transcriptomics and image-based techniques, we perceived it as the above-mentioned sentence.

4) Some statements seem to be
- exaggerated, like:

--Line 24, “MIST excelled”. Based on 1 sample, it is vague to claim this.

Thank you for pointing it out. We changed ‘excelled’ to ‘facilitated’.

--Line 246, “greatly”. The difference in Fig 3a seem not to be great, in HBCB1 from ~ 0.9 (McImpute) to ~ 0.8 (MIST); in some sample the difference is even less.

Thank you for the point. We changed “greatly” to “significantly” since the improvement is statistically significantly (line 155-162).

- overcritical, like

--Line 264, “ST data provided by 10X Visium might hinder the accuracy of ST data analysis”. I think ‘hinder’ is not the right term.

Thank you for the suggestion. We changed the phrase ‘hinder the accuracy of ST analysis’ to ‘suffer from the sparsity issue’.

--Line 49, “... none of them is able to establish the boundaries of different tissue regions.” Is this true? What is shown in Fig S2c, clusters by BayesSpace, instead?

In Fig. S2, we showed that BayesSpace assigned every spot to a region (panel C), which are not true according to the manual annotation on the left (panel A). It did not leave boundary spots out that might be composed of mixtures of different cell types from regions with different pathologies. However, MIST only assign spots to regions with high confidence, leaving spots at the boundaries out (**panel B**). This would allow other important analyses, for example, analyzing communications of different regions through decomposing the contributions of transcriptomes from functionally different regions at the boundary spots, which is one of the features we are planning to add.

--Line 238, “We view these isolated spots as important elements for domain-specific studies, such as tumor microenvironments”. To merge all spots that could not be assigned to a cluster into one functional group seems imprecise.

Sorry for not making it clear and thank you for bringing the ‘isolated’ spots up as we think they are very important. The spots that were not assigned to a cluster are not classified in one functional group. Since each spot in ST is a mixture of different cells or cell types, these isolated spots could mixtures of different cell types coming from the major detected regions. However, the mixing rate might be different for different spots, making them heterogenous and thus not assigned to a particular region. We believe there are more to be analyzed for these isolated spots, which is one of our major future directions. The discussion on this issue is added in the updated manuscript (**line 272-277**)

--Line 265, “dramatically improves the signal-to-noise ratio.” Where are the values of signal-to-noise-ratio shown?

Sorry for causing the confusion, we changed it to “increased the signals and revealed the genes’ spatial co-expression patterns”.

Reviewer #2:

Though the authors provide responses to my comments, I don't think they address my major concerns.

1) The authors only compared with simple clustering methods such as KMeans, Leiden, Louvain, and Hierarchical Clustering. These methods are not comparable with MIST, as they are not well used to cluster spatial data. I do recommend comparing with the spatial designed methods, such as STAGATE (PMID: 35365632) and spaGCN (PMID: 34711970).

Thank you for pointing these two recently published ST clustering methods. We did compare with BayesSpace, which is a recently published method leveraging spatial information in detecting major clusters within ST data.

STAGATE is a method published on 2022 April 1st, according to Dr. Ross Cloney's email, "*When evaluating your revised manuscript, we will not consider any similar papers published in the meantime to compromise the novelty of your study. See here for more information.*", we think it is unfair to compare MIST with it. MIST was submitted to Nature Communications on 06/22/2021 with the preprint version available online (<https://doi.org/10.21203/rs.3.rs-647777/v1>), and another preprint was also available at bioRxiv (<https://www.biorxiv.org/content/10.1101/2021.05.14.443446v1>) on 05/17/2021. Moreover, we found STAGATE very similar to MIST in its first step (Constructing spatial neighbor network) in Fig. 1. Both MIST and STAGATE constructs a spatial network by removing edges connecting spots with low transcriptomes' similarities. Therefore, we decided not to compare with STAGATE.

We added comparison with SpaGCN to the region detection study and updated **Fig. 2** and demonstrated that MIST have higher median silhouette coefficient over the 13 compared ST datasets.

2) Moreover, the authors show the clustering performance based on the Silhouette index, which is insufficient and inappropriate, especially when the spatial data has ground truth. Though the authors include the 10x visium data, I would recommend doing the comparisons based on the 10x Visium dataset containing spatial expressions of 12 human dorsolateral prefrontal cortex (DLPFC) section (PMID: 33558695). The comparison results are preferred to be shown as figure with ARI metrics.

The reason we used Silhouette coefficient score is because: (1) In most cases, spot-level human experts' annotations are not available. For example, 11 out of 13 samples in the original Supplementary Table S1 do not have such spot-level annotations. Therefore, it's hard to calculate other label-based metrics such as adjusted rand index (ARI); (2) Silhouette coefficient score was suggested by other reviewers, and we agree that it is fair to use this metric to evaluate the concordance between the detected regions and the underlying structures within the data.

Nonetheless, we thank the reviewer for pointing out a great resource – the Human DLPFC data sets, to evaluate MIST's region detection accuracy. As suggested by the reviewer, we calculated the adjusted rand index (ARI) score for each data set and summarized the results in supplementary figure (**Supp. Fig. 7**).

To conclude, we found that MIST's performance is not significantly different than SpaGCN (P-value=0.7) nor BayesSpace (P-value=0.2) when evaluating on MIST-detected spots (**Supp. Fig. 7**). All these three spatial information-based methods significantly outperformed other non-spatial-based methods (P-value $\leq 2 \times 10^{-4}$).

Since MIST will leave out some spots which are not connected to any detected regions due to dissimilar transcriptomic profiles, we also compared MIST's ARI on MIST-detected spots

against other methods' ARI on all spots. We found that MIST's ARI is significantly higher than other methods on the Human DLPFC data with $\Delta ARI \geq 0.17$ (paired T-test P-value = 0.01) when comparing with BayesSpace (**Supp. Fig. 7**).

More importantly, we demonstrated MIST detected spots significantly improved the ARI for each of the other methods with $\Delta ARI \geq 0.12$ (paired T-test P-value ≤ 0.02) (**Supp. Fig. 7**) by leaving out spots that are not similar to any major regions at the transcriptome level. This result shows that MIST will selectively assign regions to spots with high confidence and could help other methods for more accurate region detection.

3) The authors argued that the cell-type deconvolutions are not comparable with MIST, however, I would like to check if the imputed data by MIST does improve the deconvolution performance? It is one way to evaluate if the imputation improves downstream analysis.

Thank you for the recommendation. Yes, we agree that it would be great if we could assess if MIST imputed data improved the accuracy of deconvolution methods. However, we think this could be an individual project as the difficulties could be foreseen in (1) finding proper datasets with the ground-truth cell type proportion; (2) generating simulation datasets; (3) selecting (benchmarking) deconvolution methods; (4) comprehensive evaluation. We are currently working on utilizing MIST to help improve deconvolution results in another project, which is not the scope of the current project.

4) I would not agree with the authors' comments saying they are unable to find a method that uses spatial imputation to benchmark with MIST, since the authors show that MIST outperformed McImpute by achieving a higher Pearson Correlation Coefficient (PCC) and lower Rooted Mean Square Error (RMSE) when recovering hold-out gene expression values. Could you add DeepImpute (PMID: 31627739) and make a figure on PCC and RMSE with each method (MIST, DeepImpute, McImpute, MAGIC, SAVER)?

Thank you for suggesting. We have added deepImpute in our holdout experiments and compared the performance with MIST. We showed that although deepImpute performs reasonably well, MIST outperformed deepImpute in both PCC and RMSE metrics (**Fig. 3**).

Overall, I would recommend the authors to verify the performance of MIST at both imputation and clustering aspects, which are the two major components of this method, through meaningful comparisons.

Thank you so much for the comments and suggestions in both aspects, which made this study more rigorous and solid.

Reviewer #3 (Remarks to the Author):

I have no more comments.

Reviewer #4 (Remarks to the Author):

The authors have addressed all my concerns to my satisfaction.

Reviewers' Comments:

Reviewer #1:

Remarks to the Author:

The authors have addressed most of my comments. However, some comments were misunderstood or not taken into account. I am trying to be clearer:

1) Annotation

MIST facilitates annotation of the discovered regions by applying pathway annotation to differentially expressed genes. This type of annotation, i.e. annotation and naming, is different from annotation by a pathologist, i.e. labelling of cell types, inflammation, morphological changes and abnormalities, as mentioned in the introduction (lines 43-46).

If the authors are thinking of annotating regions similar to pathologist annotations, e.g. cell types and disease states, as indicated in Fig. 2a and SFig 2.1, then this should be demonstrated.

Comparing MIST with tools and the regions they identify, as was done in SFig 2.2-2.12, is in my opinion insufficient in this case; all tools can agree in the regions they identify and still can be completely different from the regions a pathologist has annotated. Pathologist annotations such as those available for the 12 prostate cancer samples, one of which the authors already use (Berglund et al., 2018), added to a comparison of regions identified by MIST and other tools, would be more helpful.

If a more general annotation, i.e. commenting and naming, of regions is meant instead, the comparison as shown in SFig 2.2-2.12 is sufficient, however, it should be stated clearer in the manuscript.

Also, the manuscript still mentions that MIST automatically annotates regions.

2) Gene expression profiles

It is not clearly defined, but gene expression profile is often understood as a vector with a single value per gene for the genes genome-wide. MIST provides a gene expression matrix of the discovered regions (one value per gene and spot within a region genome-wide). This should be clarified in the manuscript.

3) In Figure 2d, it is still not clear which two cell clusters are being compared in each volcano plot. Which cell cluster is the basis for the up-regulated genes on the left (highlighted in blue if significant) and which cell cluster is the basis for the up-regulated genes on the right (highlighted in red if significant) of each volcano? In other terms, what is control and what is condition in each volcano plot? In addition, in the method section, an adjusted p-value cut-off of $10e^{-5}$ and a log₂fc cut-off of 0.5 are mentioned but in the volcano plot an adjusted p-value cut-off of 0.01 and no fc cut-off are marked.

Minor comments

Line 53, the assumption that the low sensitivity value is caused by "excessive zero values" in the ST data is somewhat hypothetical in my eyes and needs a better reference if true.

Line 49, what the authors write in the manuscript and what they give as explanation in the reviewers' response are different things. BayesSpace, for example, is indeed able to determine the boundaries of different regions, but accepts that the confidence in spot assignment varies within a region. This should be corrected in the manuscript, as explained by the authors in the reviewer's response. The fact that spots remain unassigned to a region can also be seen as disadvantageous.

Fig. 2d: Either Gene Ontology terms may not be the best choice to show that MIST facilitates the annotation of regions when it is impossible to infer which disease-related or abnormal processes are found in a MIST region or the submitted genes are not meaningful enough. The example is therefore only limited evidence of how MIST facilitates annotation.

In addition, the full list of significantly activated pathways for each comparison should be included in the supplementary file, as it can provide important insights into the activated/deactivated processes in a region.

There are also other databases that provide enrichment of processes that can be associated with specific diseases and cell types.

Some gene names and overlap values in the supplementary files are formatted as dates.

Reviewer #2:

Remarks to the Author:

I have no more concerns regarding this manuscript.

All four reviewers provided very thoughtful reviews and constructive suggestions, which we appreciate and have accommodated to the best of our ability. Below, for the sake of convenience, we italicize the original reviewer comments, and place our responses in regular roman font.

Reviewer #1 (Remarks to the Author):

The authors have addressed most of my comments. However, some comments were misunderstood or not taken into account. I am trying to be clearer:

1) Annotation

MIST facilitates annotation of the discovered regions by applying pathway annotation to differentially expressed genes. This type of annotation, i.e. annotation and naming, is different from annotation by a pathologist, i.e. labelling of cell types, inflammation, morphological changes and abnormalities, as mentioned in the introduction (lines 43-46).

If the authors are thinking of annotating regions similar to pathologist annotations, e.g. cell types and disease states, as indicated in Fig. 2a and SFig 2.1, then this should be demonstrated. Comparing MIST with tools and the regions they identify, as was done in SFig 2.2-2.12, is in my opinion insufficient in this case; All tools can agree in the regions they identify and still can be completely different from the regions a pathologist has annotated. Pathologist annotations such as those available for the 12 prostate cancer samples, one of which the authors already use (Berglund et al., 2018), added to a comparison of regions identified by MIST and other tools, would be more helpful. If a more general annotation, i.e. commenting and naming, of regions is meant instead, the comparison as shown in SFIGs 2.2-2.12 is sufficient, however, it should be stated clearer in the manuscript.

We do agree with the reviewer that “*This type of annotation, i.e. annotation and naming, is different from annotation by a pathologist*”. To ensure that there is no inaccurate comparison, we removed all references to a pathologist in the text. In addition, we defined regions detected by MIST as **molecular regions** (line 19, 63, 70 etc.) because MIST identifies regions based on the similarity of the molecular profiles across the spatial spots. To understand the molecular functions of these regions, we use differential gene analysis followed by gene set enrichment analysis (GSEA). We agree that this is fundamentally different from how pathologists annotate histological images such as H&E stains.

Molecular regions identified by MIST, based on transcriptome similarity, agree with annotations based on cell morphology, as demonstrated in **Fig. 2a-b**. Following Reviewer 1’s suggestion, we compared MIST-identified regions with those identified by other tools (**SFig 2.2-12**). However, it is challenging to do the same as demonstrated in **Fig. 2a** as there is a lack of manual annotation in most of the original data sources. To address this limitation, we computed the Silhouette coefficient, a metric for the goodness of clustering. MIST outperformed most of the approaches in the median value (**Fig. 2g**).

Also, the manuscript still mentions that MIST automatically annotates regions.

Thank you for catching this mistake. We replaced “Automatically annotating regions for a melanoma sample” with “MIST facilitates annotations of molecular regions in melanoma” (line

93).

2) Gene expression profiles

It is not clearly defined, but gene expression profile is often understood as a vector with a single value per gene for the genes genome-wide. MIST provides a gene expression matrix of the discovered regions (one value per gene and spot within a region genome-wide). This should be clarified in the manuscript.

We thank the reviewer for the suggestion. We clarified this point by adding the following sentence: “By doing so, MIST provides a region-specific and spot-level expression profile for the whole transcriptome, enabling an understanding of regional transcriptional differences.” (line 78-79).

3) In Figure 2d, it is still not clear which two cell clusters are being compared in each volcano plot. Which cell cluster is the basis for the up-regulated genes on the left (highlighted in blue if significant) and which cell cluster is the basis for the up-regulated genes on the right (highlighted in red if significant) of each volcano? In other terms, what is control and what is condition in each volcano plot?

We apologize for poor presentation of this figure. We revised the legend for Figure 2d as follows: “Volcano plot of significantly enriched (red dots) and repressed (blue dots) genes in each detected region compared against spots from all other regions, with the top 10 significantly activated genes listed on the margin. Panels from left to right represent different regions: black (tumor), dark orange: (lymphoid) and red (stroma).” (line 622-625). In other words, the control for each region is spots from all other regions.

In addition, in the method section, an adjusted p-value cut-off of $10e-5$ and a $\log_{2}fc$ cut-off of 0.5 are mentioned but in the volcano plot an adjusted p-value cut-off of 0.01 and no fc cut-off are marked.

Thank you for highlighting the inconsistency of the adjusted p-value and fold-change cutoffs in the **Methods**. We double checked our code and confirmed that we used the same adjusted p-value of 0.01 and fold-change cutoff of 50%. We corrected it as follows: “To get regional differentially expressed genes, we compared the spots within the region against spots from all other regions. Then, we selected genes with a fold change greater than 50% and adjusted P-value less than 10^{-2} .” (line 456 to line 459)

Minor comments

Line 53, the assumption that the low sensitivity value is caused by "excessive zero values" in the ST data is somewhat hypothetical in my eyes and needs a better reference if true.

To address the Reviewer’s concern, we plotted the sparsity levels of all 13 datasets listed in the **Supp. Table 1** in the following violin plot (also **Supp. Figure 1**). The figure shows that 8 out of 13 samples have median gene-level sparsity of over 80% and all samples have median gene-level sparsity of over 60%.

Figure 1 | Gene-level sparsity for each data set in Supp. Table 1. X-axis indicates sample name. Y-axis shows the distribution of the percentage of spots having zero values for each gene. Each violin plot includes a boxplot with median value shown as a white dot.

Line 49, what the authors write in the manuscript and what they give as explanation in the reviewers' response are different things. BayesSpace, for example, is indeed able to determine the boundaries of different regions, but accepts that the confidence in spot assignment varies within a region. This should be corrected in the manuscript, as explained by the authors in the reviewer's response.

Per Reviewer's suggestions, we changed the original line 49 to current line 48-54 to clarify the difference between MIST and other spatial clustering algorithms, such as BayesSpace. The major difference in MIST is that it excludes spots that are not similar to any molecular regions, likely caused by multiple cells from different regions included in one spot. Here is what we wrote: "Some computational methods have sought to assign every spot to a cluster, even when that spot resides at the boundary between pathogenically different regions. However, these spots at the boundaries might be cell type admixtures due to the lack of single-cell resolution of spatial spots, thus not belonging to a specific molecular region and should be excluded from cluster-based analysis in the downstream. Unbiased molecular region mapping within ST tissues that excludes potential admixture spots would therefore present a substantial improvement in accuracy of ST data interpretation." (line 48-54).

The fact that spots remain unassigned to a region can also be seen as disadvantageous.

We discussed this point as a potential limitation and our plan to analyze these unassigned spots in future studies in line 283-288.

Fig. 2d: Either Gene Ontology terms may not be the best choice to show that MIST facilitates the annotation of regions when it is impossible to infer which disease-related or abnormal processes are found in a MIST region or the submitted genes are not meaningful enough. The example is therefore only limited evidence of how MIST facilitates annotation. There are also other

databases that provide enrichment of processes that can be associated with specific diseases and cell types.

Thank you for the comment and suggestion. We used the *GSEApY* interface (<https://gseapy.readthedocs.io/en/latest/introduction.html>), which allows users to test the significance of overlap between user-identified regional DEGs and up to 201 available gene sets, such as the Cancer Cell Line Encyclopedia.

We do agree that gene ontology terms might not be the best gene sets to demonstrate MIST's efficacy in **Figure 2e**. Therefore, we additionally compared regional DEGs with over 1000 cancer related gene sets from the Cancer Cell Line Encyclopedia (CCLE) database by Broad Institute (Ghandi, Mahmoud et. al., 2019). We added the result on the melanoma sample in **Fig. 2f**. With the CCLE database, we showed that the black region is enriched with skin cancer related cell lines. We also show the analyses on other cancer samples in the **Supp. Fig. 2** and found that the performance is reliably generalizable across samples. We included the GSEA results on CCLE database in **Supp. Table**.

References used in this reply:

Ghandi, Mahmoud, et al. "Next-generation characterization of the cancer cell line encyclopedia." *Nature* 569.7757 (2019): 503-508.

In addition, the full list of significantly activated pathways for each comparison should be included in the supplementary file, as it can provide important insights into the activated/deactivated processes in a region.

We provided the enriched GO terms and CCLE cell lines in supplementary files.

Some gene names and overlap values in the supplementary files are formatted as dates.

We are sorry for this incorrect formatting caused by saving comma-delimited files into excel files. We corrected such errors and provided the updated DEG results in supplementary files.